# Algal cell bionics as a step towards photosynthesis-independent hydrogen production

Zhijun Xu[1], Jiarui Qi[1], Shengliang Wang[1], Xiaoman Liu[1], Mei Li [2], Stephen Mann [2,3,4] ✉ & Xin Huang [1] ✉

The engineering and modulation of living micro-organisms is a key challenge in green bio-manufacturing for the development of sustainable and carbon-neutral energy technologies. Here, we develop a cellular bionic approach in which living algal cells are interfaced with an ultra-thin shell of a conductive polymer along with a calcium carbonate exoskeleton to produce a discrete cellular micro-niche capable of sustained photosynthetic and photosynthetic-independent hydrogen production. The surface-augmented algal cells induce oxygen depletion, conduct photo-induced extracellular electrons, and provide structural and chemical stability that collectively give rise to localized hypoxic conditions and concomitant hydrogenase activity under daylight in air. We show that assembly of the living cellular micro-niche opens a direct extra-cellular photoelectron pathway to hydrogenase resulting in photosynthesis-independent hydrogen evolution for 200 d. In addition, surface-conductive dead algal cells continue to produce hydrogen for up to 8 d due to their structural stability and retention of functional hydrogenases. Overall, the integration of artificial biological hydrogen production pathways and natural photosynthesis in surface-augmented algal cells provides a cellular bionic approach to enhanced green hydrogen production under environmentally benign conditions and could pave the way to new opportunities in sustainable energy production.

It has been widely recognized that the immoderate consumption of fossil fuels has caused serious environmental problems such as global warming and climate change, which may greatly restrict the economic and social development of future human societies. The utilization of renewable and clean energy resources is highly desirable to address these global problems[1-4]. In this regard, the development of efficient green bio-manufacturing is undoubtedly needed to address global energy and environmental problems[5-7].

In particular, engineering of algal cells for $H_2$ production, which started as early as 1942[8], offers a potential strategy based on the utilization of electrons from light-capturing photosystems to drive hydrogenase activity at the reducing end of the photosynthetic electron transfer chain[9]. However, photohydrogen generation is transient and often occurs for only a few minutes during dark-light transitions since the accompanying photosynthetically generated oxygen strongly attenuates hydrogenase

[1]MIIT Key Laboratory of Critical Materials Technology for New Energy Conversion and Storage, School of Chemistry and Chemical Engineering, Harbin Institute of Technology, 150001 Harbin, Heilongjiang, China. [2]Max Planck Bristol Centre for Minimal Biology, Centre for Protolife Research and Centre for Organized Matter Chemistry, School of Chemistry, University of Bristol, Bristol BS8 1TS, UK. [3]School of Materials Science and Engineering, Shanghai Jiao Tong University, 200240 Shanghai, People's Republic of China. [4]Zhangjiang Institute for Advanced Study (ZIAS), Shanghai Jiao Tong University, 429 Zhangheng Road, 201203 Shanghai, People's Republic of China. ✉e-mail: s.mann@bristol.ac.uk; xinhuang@hit.edu.cn

activity. Consequently, strategies based on anaerobic fermentation, respiration enhancement, nutrient deprivation and gene engineering of oxygen-tolerant hydrogenases have been developed to eliminate the negative impact of oxygen production[10–17]. In addition, recent studies have demonstrated the feasibility of inducing algal cell aggregation as a strategy for creating localized hypoxic conditions in air[18–22]. For example, the spatial organization and aggregation of algal and bacterial cells within dextran-in-polyethylene glycol aqueous liquid-liquid phase-separated microdroplets produced photobiological hydrogen at the rate of 0.44 µmol $H_2$ (mg chlorophyll)$^{-1}$ h$^{-1}$ for 168 h in air[21]. Alternatively, photocatalytic models for solar conversion based on inorganic semiconductor/hydrogenase complexes[23,24] or inorganic semiconductor/bacterial cell hybrid systems[25–31] have been developed as promising approaches to hydrogen production. For instance, in situ formation of CdS nanoparticles on the surface of engineered *Escherichia coli* under aerobic conditions[25] gave rise to continuous hydrogen production over a period of 96 h. The activity was considerably prolonged compared with the strategies such as gene engineering. Despite the above developments, both the relatively low hydrogen production rates and limited durations of hydrogen production, typically from a few hours to several hundred hours, remain major challenges and bottlenecks for future advances in photobiological hydrogen production. Recently, mesoporous microspheres[32], platinum nanoclusters[33], and polypyrole[34] have been employed to fabricate biohybrids capable of cell-free hydrogen production, tunable electrical conductivity or self-enhancing photoactivity. In addition, construction of bionic functional layers around living cells with ultra-thin shells of a coacervate phase[35], semiconductors and carbon nanotubes[36], polypyrole[37] or calcium carbonate[38] has proved to be an effective strategy for enhancing natural cellular properties and augmenting living organisms with non-natural functions.

Inspired by these approaches, herein we develop a methodology to augment the hydrogen producing capability of the photosynthetic organism *Chlorella pyrenoidosa* by enhancing and prolonging hydrogenase activity in air. To achieve this, we envelop live algal cells in concentric layers of a conductive organic polymer and biocompatible inorganic mineral to produce a discrete cellular bionic micro-niche that supports oxygen consuming reactions, facilitates transport of extracellular electrons, provides structural stability in both live and dead cells, and prevents the acidification of the environment. Specifically, the abiotic shell comprises an ultra-thin Fe(III)-doped polypyrole (PPy) inner layer and outer exoskeleton of calcium carbonate (CaCO$_3$). Entrapment of Fe(III) in the polymer layer depletes the dissolved oxygen concentration via Fe(III)/O$_2$ catalytic oxidation of added ascorbate to produce sustainable hypoxic conditions within 12 h, which in turn activates hydrogenase activity and initiates photobiological hydrogen production. PPy also acts as a conductive media for the capture and transport of extracellular electrons into the algal cells thereby enhancing the rate of hydrogen production. Collectively, these processes facilitate metabolic switching from oxygen production to hydrogen evolution in air, which is sustained for up to 200 d in the live surface-augmented algal cells. We also show that PPy/CaCO$_3$-coated dead algal cells maintain their hydrogenase/hydrogen activity for up to 8 d in the presence of a photosensitizer and sacrificial electron donor due to structural and functional integrity of the augmented cell wall. Taken together, our work demonstrates a cellular bionics approach to enhanced and prolonged green hydrogen production in which the surface-augmentation of algal cells facilitates the integration of natural photosynthetic pathways and artificial routes to hydrogenase activation. Future development of this proof-of-principle approach could pave the way to new opportunities in sustainable energy production.

## Results

### Micro-niche construction by polymer/inorganic coating of live algal cells

Cellular micro-niches geared to photosynthetic hydrogen production were constructed by coating individual live cells of *Chlorella pyrenoidosa* with a concentrically arranged bilayered shell of PPy and CaCO$_3$ using a two-step process (Fig. 1a). A PPy shell was initially assembled on the algal cell wall by culturing with iron (III) chloride for 30 min followed by addition of pyrrole. Binding of Fe (III) to the algal cells (Supplementary Fig. 1) triggered the oxidative polymerization of pyrrole specifically on the cell wall to produce PPy-coated living cells that remained dispersed in water. Scanning electron microscopy (SEM) and transmission electron microscopy (TEM) images showed a continuous interconnected PPy layer ~5–10 nm in thickness, which tightly enclosed and infiltrated the cell wall (Fig. 1b, c and Supplementary Figs. 2a, b and 3). Successful cellular attachment of PPy was confirmed by UV/visible spectroscopy (Fig. 1e) and cyclic voltammetry (CV) (Supplementary Fig. 4), which showed a broad absorption peak between 400 and 490 nm and an increased current density, respectively.

Having assembled the PPy layer, aqueous calcium chloride and sodium carbonate were added to the suspension of polymer-coated cells to induce the nucleation and growth of an integrated CaCO$_3$ overlayer (Fig. 1d and Supplementary Fig. 5). Formation of the exoskeleton specifically on the surface of the negatively charged PPy-coated cells was attributed to template-mediated biomineralization arising from the electrostatic binding of Ca$^{2+}$ ions to the polymer-decorated algal cell wall (Supplementary Fig. 6). X-ray powder diffraction (XRD) patterns confirmed that the CaCO$_3$ overlayer comprised a mixture of crystalline calcite and vaterite (Fig. 1f), consistent with Fourier transform infrared (FTIR) spectra that exhibited strong peaks at 745, 877, 1407, and 1460 cm$^{-1}$ (Supplementary Fig. 7). Doping the CaCO$_3$ phase with a fluorophore during the biomineralization process gave rise to continuous green fluorescence around each *Chlorella* cell, indicating that the inorganic exoskeleton was firmly attached and enclosed the entire cell periphery (Fig. 1h–j).

Integration of the two-tiered shell architecture with the algal cell wall was confirmed by X-ray photoelectron spectroscopy (XPS) that showed high intensity peaks corresponding to characteristic Ca and Fe binding energies (Supplementary Fig. 8). Nitrogen adsorption-desorption isotherms indicated that the PPy/CaCO$_3$ hybrid layer was mesoporous with an average pore size of *ca.* 3.92 nm (Supplementary Fig. 9), suggesting that nutrients such as metal ions in the external media remained accessible to the live *Chlorella* cells after coating of the cell wall. AFM measurements indicated that the coated algal cells were approximately eight-times stiffer than the native cell wall (Young's moduli, 8.53 and 1.06 GPa, respectively) (Fig. 1g). Consequently, the augmented cell wall remained essentially intact even in dead PPy/CaCO$_3$-coated *Chlorella* cells obtained from cultures maintained for 7 months (Supplementary Fig. 10). Cell viability studies indicated that the polymer/inorganic shell was biocompatible with over 96% of the engineered *Chlorella* cells remaining active (Supplementary Figs. 11 and 12). Moreover, cell viability of the coated algal cells increased almost two-fold compared with the native cells when exposed to silver nanoparticles or elevated temperatures (60 °C) for 60 min (Supplementary Fig. 13), indicating that the PPy/CaCO$_3$ shell provided some measure of protection under adverse conditions. Significantly, determination of the maximum quantum yield of photosystem II (PSII) in a dark-adapted state using chlorophyll fluorescence measurements showed that algal photosynthetic activity was essentially unchanged over 7 d after coating of the cell wall (Supplementary Fig. 14). This was consistent with the well-maintained chlorophyll content of the PPy/CaCO$_3$-engineered cells observed over the same period (Supplementary Fig. 15).

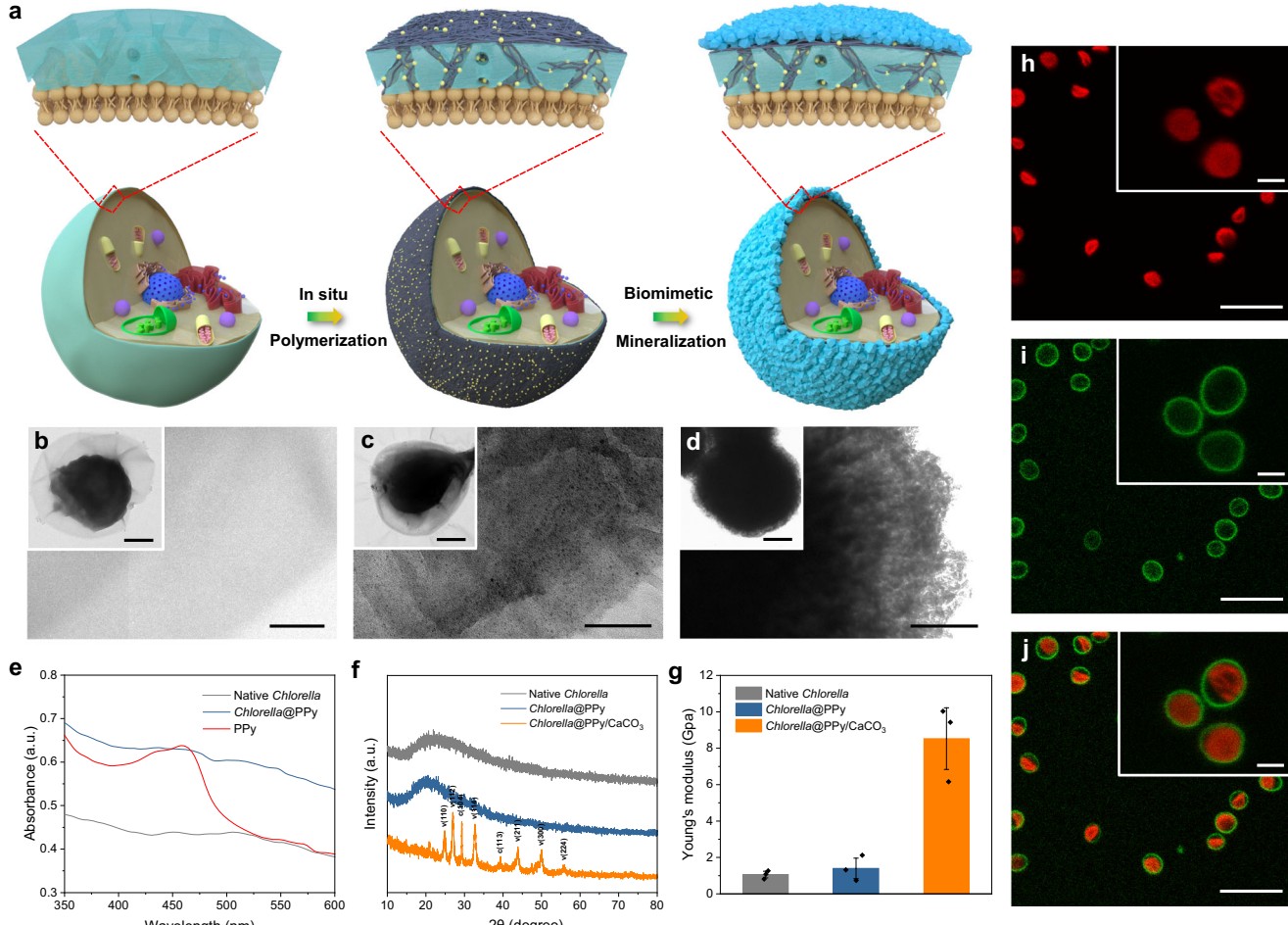

**Fig. 1 | Micro-niche construction by polymer/inorganic coating of live algal cells. a** Schematic illustration showing the stepwise construction of concentrically arranged PPy and CaCO$_3$ thin shells on the cell wall of the alga *Chlorella pyrenoidosa* (small yellow spheres, Fe(III) ions; black inner shell, PPy; light blue outer shell, CaCO$_3$). **b–d** TEM images of localized regions and whole cells (insets) of native (**b**), PPy-coated (**c**) and PPy/CaCO$_3$-coated *Chlorella* cells (**d**) showing increased electron density and granularity associated with the coatings. **e** UV–Vis spectra of PPy, native and PPy-coated *Chlorella* cells. **f** XRD patterns of native, PPy-coated and PPy/CaCO$_3$-coated algal cells; Miller indices for calcite (*c*) and vaterite (*v*) are shown.

**g** Plot of Young's modulus of native, PPy-coated and PPy/CaCO$_3$-coated *Chlorella* cell walls. Data are presented as mean values ± SD, error bars indicate standard deviations (*n* = 3, biologically independent samples). **h, i** Confocal fluorescence microscope images of PPy/CaCO$_3$-coated *Chlorella* cells displayed in red (**h**) and green (**i**) channels. **j** overlay image of **h** and **i**. Red fluorescence, intracellular chlorophyll; green fluorescence, FITC-labeled CaCO$_3$ shell. Scale bars: (**b**), 100 and 500 nm (inset); (**c**), 100 nm and 1 μm (inset); (**d**), 200 nm and 1 μm (inset); **h–j** 10 and 2 μm (insets). All relevant experiments were performed independently at least three times with similar results. Source data are provided as a Source Data file.

## Hypoxic photosynthetic hydrogen production in surface-augmented algal cells

We sought to utilize the PPy/CaCO$_3$-coating methodology as a strategy to generate living cell micro-niches capable of prolonged photosynthetic hydrogen production in air. As algal cells metabolically produce oxygen via the oxygen-evolving complex of PSII and exhibit negligible hydrogen generation due to inhibited expression and activation of hydrogenases in air, we used the integrated Fe(III)-doped PPy and CaCO$_3$ layers to generate prolonged hypoxic conditions (Fig. 2a). To achieve this, we added sodium ascorbate to the cell cultures and monitored the effect of Fe(III)-catalyzed oxidation at the cell wall on the dissolved oxygen concentration (Supplementary Note 1). The PPy- and PPy/CaCO$_3$-coated algal cells, but not the native *Chlorella* cells, showed a marked depletion in dissolved oxygen concentration, typically from *ca.* 5 to 0.2 mg/L over 12 h (Fig. 2b). Consequently, the [FeFe]-hydrogenase was successfully expressed and activated (Supplementary Fig. 16), and within 2 d the coated algal cells switched from photosynthetic oxygen to hydrogen generation under daylight (Supplementary Fig. 17), which was maintained typically for 14 d (Fig. 2c). During this period, oxidation of the substrate (ascorbate) within the

native and PPy-coated algal cells gave rise to a decrease in pH from 7.2 to *ca.* 6.0 (Fig. 2d). In contrast, the pH only decreased to *ca.* 6.9 for the PPy/CaCO$_3$-coated cells due to buffering by the inorganic CaCO$_3$ exoskeleton (Fig. 2d).

As photoelectrons for *Chlorella* cell-derived hydrogen production originate mainly from PSII, which captures solar energy and drives electron transfer in the thylakoid membrane (Fig. 2a), we determined the maximum potential quantum efficiency of PSII in the native and coated algal cells to evaluate the capability of the hybrid systems for extended hydrogen production. As expected, all the algal cell-based systems showed decreases in photosynthetic activity under hypoxic conditions ($F_v/F_m$ = 0.4–0.54; $F_v$, variable fluorescence; $F_m$, maximum fluorescence) compared with the measurements recorded under aerobic conditions ($F_v/F_m$ = 0.6–0.7, Supplementary Fig. 14). This was consistent with the observed slow time-dependent increase in chlorophyll content of the PPy/CaCO$_3$-coated algal cells (Supplementary Fig. 15). Nevertheless, the PPy/CaCO$_3$-coated cells retained 72% of their original PSII activity over 6 d, with corresponding values of 54% and 57% for the native and PPy-coated cells, respectively (Fig. 2e). As hydrogenases are inhibited under acidic conditions, we speculated

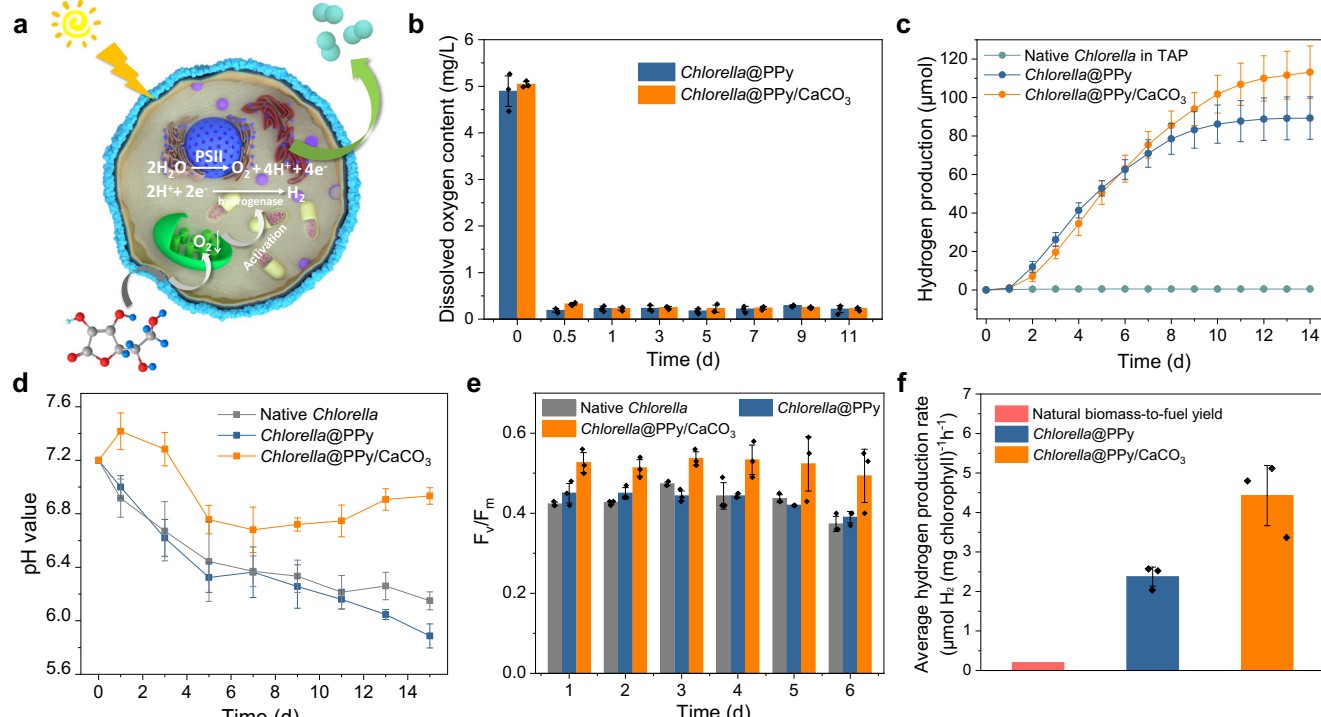

**Fig. 2 | Hypoxic photosynthetic hydrogen production in surface-augmented algal cells. a** Schematic illustration of photosynthesis in PPy/CaCO₃-coated *Chlorella* cells. Hypoxic conditions are generated by depletion of oxygen levels due to Fe(III)/PPy-mediated oxidation of ascorbate (shown as molecular graphic) present in the external environment, giving rise to hydrogenase activation and hydrogen production (paired cyan circles). The decrease of pH arises from the oxidation of added ascorbate and is buffered by the CaCO₃ shell, which also facilitates prolonged hydrogenase activity. **b**–**e** Time-dependent measurements of dissolved oxygen concentration (**b**), hydrogen production (**c**), pH (**d**) and photosynthetic activity ($F_v/F_m$) (**e**) in an ascorbate-containing TAP culture medium in the presence of native, PPy-coated or PPy/CaCO₃-coated *Chlorella* cells. Data are presented as mean values ± SD, error bars indicate standard deviations ($n = 3$,

biologically independent samples). Both the PPy- and PPy/CaCO₃-coated algal cells generate hypoxic conditions and photosynthetically produce hydrogen (**b**, **c**). The CaCO₃ exoskeleton acts as a pH buffer against acidification of the environment (**d**) and enhances hydrogen production (**c**). **f** Plots of average production rate for the natural instantaneous biomass-to-fuel yield, PPy-coated cells and PPy/CaCO₃-coated algal cells. The production rates are determined between time points of 2 and 7 d. Data are presented as mean values ± SD, error bars indicate standard deviations ($n = 3$, biologically independent samples). All Samples are cultivated in seal vials with sodium ascorbate-containing TAP culture medium and exposed to daylight with an intensity of 65 μE m⁻² s⁻¹. All relevant experiments are performed independently at least three times with similar results. Source data are provided as a Source Data file.

that the higher photosynthetic activity of hydrogen production observed in the presence of the outer CaCO₃ exoskeleton was due to buffering of protons in the solution (Fig. 2d).

Taken together, our results indicate that the surface-augmentation of live algal cells with a two-layered concentric arrangement of inner and outer shells of Fe(III)-doped PPy and CaCO₃, respectively, provides a cellular micro-niche to facilitate prolonged photosynthetic hydrogenase activity. As a consequence, *Chlorella* cells coated in PPy and CaCO₃ showed a hydrogen production rate of 4.4 μmol H₂ (mg chlorophyll)⁻¹ h⁻¹, which was over 22-times higher than determined for the instantaneous biomass-to-fuel yield observed in Nature, and at least 10-times higher than the typically reported in the strategies such as aggregation[18,21] or enzymatic catalysis[19,22] (Fig. 2f and Supplementary Table 1).

**Enhanced extracellular photoelectron transport and hydrogen production in surface-conductive algal cells**

In general, increasing the number of photoelectrons transferring to hydrogenase through the photosynthetic pathway will enhance hydrogen production, but in practice this is often attenuated by hydrogenase deactivation due to concurrent photosynthetic oxygen evolution. We sought to circumvent this dilemma by employing an extracellular photoelectron pathway using an extraneous photosensitizer and sacrificial electron donor, along with enhanced cell wall conductivity to boost biological hydrogen production under hypoxic

conditions (Fig. 3a). To achieve this, we first measured the surface current associated with the native and PPy-coated *Chlorella* cells; the data indicated that embalming the algal cells in a PPy shell gave rise to a 14-times increase in surface conductivity compared to the uncoated cells (Fig. 3b–e). Based on these observations, we cultured the PPy/CaCO₃-coated *Chlorella* cells in the presence of extracellular eosin Y (EY, photosensitizer) and extracellular triethanolamine (TEOA, sacrificial electron donor) (Supplementary Fig. 18), which are known to produce photoelectrons under visible light[39]. As a result, a further 46% enhancement in the hydrogen production of the PPy/CaCO₃-coated cells was achieved under daylight, with 160 μmol hydrogen produced within 12 d compared to 110 μmol in the absence of EY and TEOA (Fig. 3f), to give a light-to-hydrogen conversion efficiency up to 0.45% (Supplementary Note 2). Negligible enhancements in hydrogen production were observed for uncoated algal cells in the presence of EY and TEOA (Supplementary Fig. 19), confirming the slow transfer of photoelectrons for the native *Chlorella* cells.

To confirm the role of extracellular electrons in augmented hypoxic photosynthesis, we measured chlorophyll fluorescence transient curves and fluorescence kinetic parameters to evaluate the absorption and capture of light energy by PSII, as well as the subsequent photosynthetic electron transfer processes for PPy/CaCO₃-coated cells in the presence or absence of EY and TEOA (Fig. 4a). For the PPy/CaCO₃-coated cells under the addition of EY and TEOA to the extracellular medium, $F_m$ in the chlorophyll fluorescence curve was

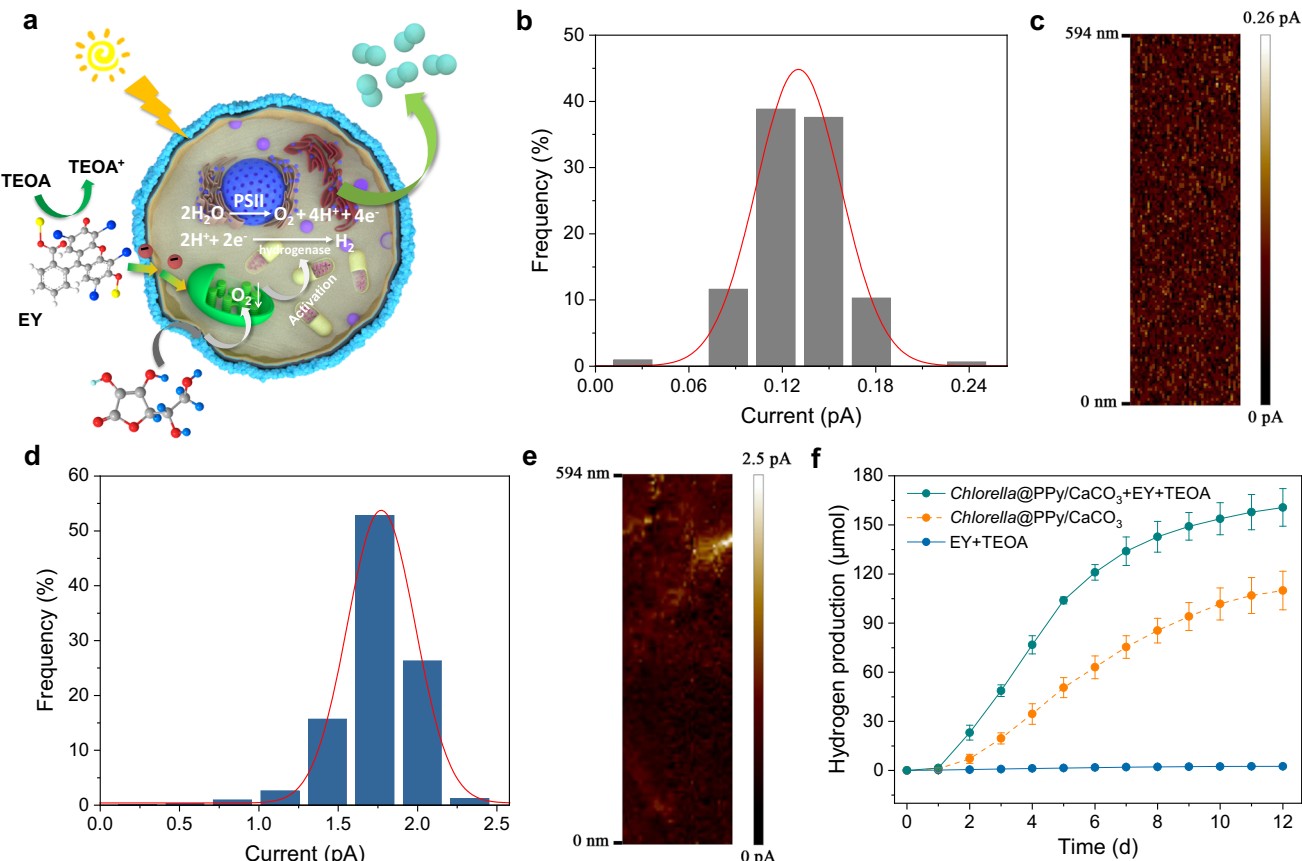

**Fig. 3 | Enhanced extracellular photoelectron transport and hydrogen production in surface-conductive algal cells. a** Schematic illustration of the PPy shell-mediated capture and translocation of artificially generated extracellular electrons for enhanced hydrogen production (paired cyan circles) in surface-coated algal cells under air; molecular graphics for EY, TEOA and ascorbate are shown. **b, c** Histogram (**b**) of measured surface current for native *Chlorella* cells with the corresponding value-distribution image (**c**). **d, e** Histogram (**d**) of measured surface current for PPy-coated *Chlorella* cells with the corresponding value-distribution image (**e**). The single peak fitting is performed using a Gaussian curve in the histograms. **f** Time-dependent measurements of hydrogen production for PPy/CaCO$_3$-coated *Chlorella* cells with or without EY and TEOA. EY and TEOA are added at the time point of 1 d. Samples are cultivated in seal vials with sodium ascorbate-containing TAP culture medium and exposed to daylight with an intensity of 65 μE m$^{-2}$ s$^{-1}$. Data are presented as mean values ± SD, error bars indicate standard deviations ($n = 3$, biologically independent samples). All relevant experiments were performed independently at least three times with similar results. Source data are provided as a Source Data file.

increased, which suggested that the activity of D1 protein in PSII was enhanced, and thus contributed to the higher efficiency of PSII electron acceptors (Fig. 4a). Moreover, an elevated quantum yield for electron transport ($\varphi_{Eo}$) was observed, indicating that the captured photoenergy from PSII was more efficiently utilized for subsequent transfer (Fig. 4b). Combining this with the enhanced $S_m$ value (energy required to completely reduce $Q_A$), it was suggested that the PQ pool was enlarged and more electrons were transferred throughout the photosynthetic chain (Fig. 4b). In addition, the density of the PSII reaction centers was also increased, as indicated by the improved number of PSII reducing centers per $CS_m$ ($RC/CS_m$) (Fig. 4b, Supplementary Table 2, and Supplementary Note 3). In terms of the specific energy flux in the photosystems, for the PPy/CaCO$_3$-coated cells in the presence of EY and TEOA, the adsorbed energy ($ABS$) per excited cross-section ($CS_m$) ($ABS/CS_m$) and trapped energy ($TR_o$) per $CS_m$ ($TR_o/CS_m$) were enhanced (Fig. 4b, Supplementary Table 2, and Supplementary Note 3), which suggested that more photoenergy was absorbed by chlorophyll and then utilized for the reduction of $Q_A$. Furthermore, the energy used for both electron transfer ($ET_o$) per $CS_m$ ($ET_o/CS_m$) and reducing end electron acceptors ($RE_o$) per $CS_m$ ($RE_o/CS_m$) were increased (Fig. 4b, Supplementary Table 2, and Supplementary Note 3), indicating that the reoxidation of the reduced $Q_A$ along with the electron transport was improved and more electrons reached the end of electron transferring chain. These observations revealed that

the extracellular electrons participated in the photosynthesis pathway and improved all the efficiencies of photoenergy absorption, capture, and transfer in the photosynthetic chain of *Chlorella* cells. Consequently, the performance indices based on absorption ($PI_{abs}$), cross-section ($PI_{cs}$) and energy conversion ($PI_{total}$), and $F_v/F_m$ values were all considerably improved (Fig. 4b, Supplementary Table 2, and Supplementary Note 3), indicative of successful internalization of extracellular electrons by the PPy/CaCO$_3$-coated *Chlorella* cells and an increase in photoactivity.

Photochemical and non-photochemical processes were also investigated by measuring the modulated chlorophyll fluorescence associated with the PPy/CaCO$_3$-coated cells in the presence or absence of EY and TEOA. As shown in Fig. 4c, the photochemical quenching coefficient ($qP$) was increased by 2.2% while the non-photochemical quenching coefficient ($qNP$) decreased by 45.6% in the presence of EY and TEOA, which suggested that more adsorbed energy was transferred to the photochemical apparatus rather than lost in the form of heat. In addition, an accelerated electron transfer rate (ETR) (by 28.6%) was also observed, indicating that more photoelectrons reached the end of the photosynthetic chain. As a result, the ratio of NADPH to NADP$^+$ as well as the ATP content were increased (Fig. 4d). Besides, the enhanced photosynthetic processing associated with the PPy/CaCO$_3$-coated cells when incubated with EY and TEOA gave rise to increased ribulose-1,5-bisphosphate

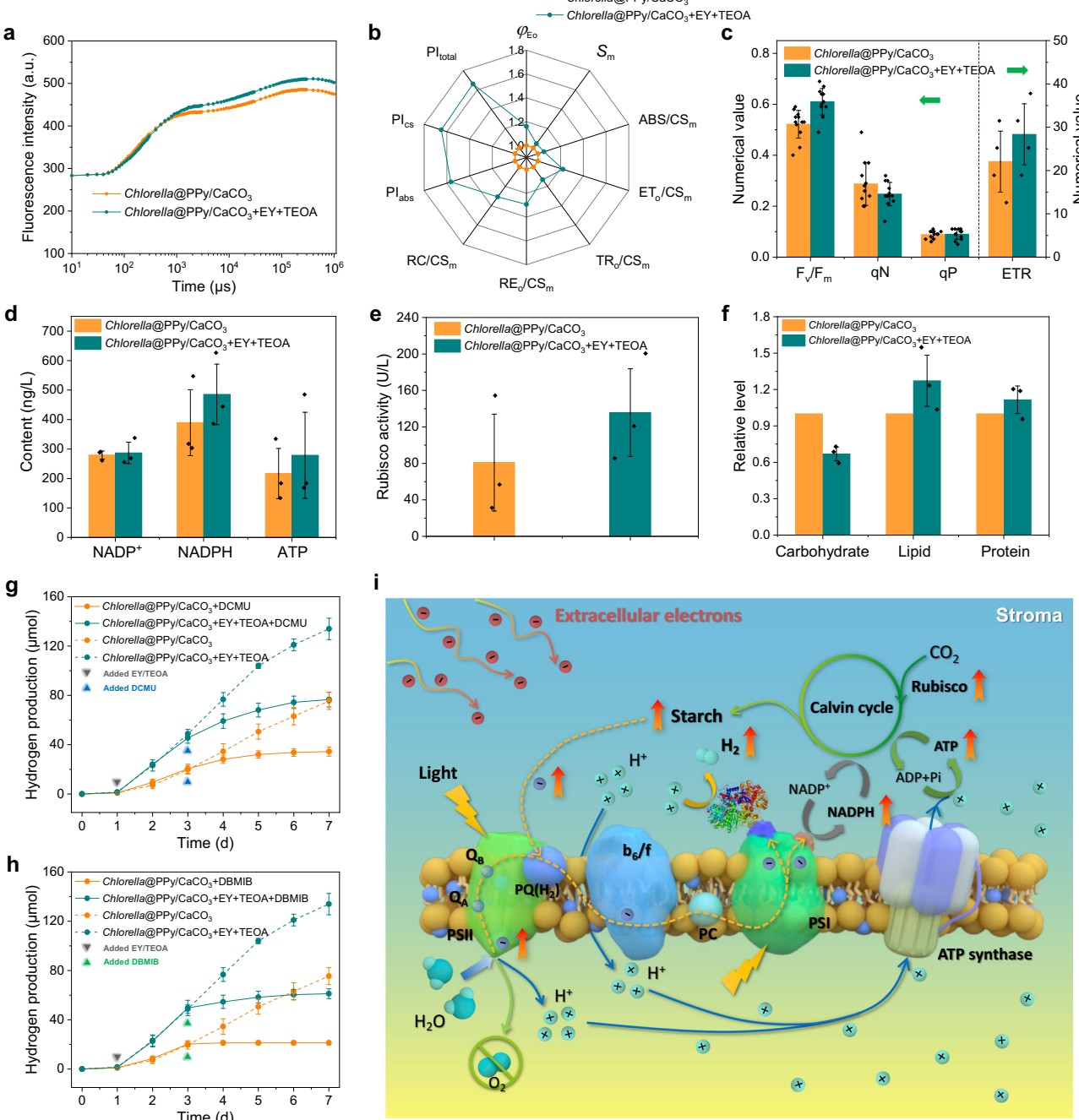

**Fig. 4 | Role of extracellular electrons in surface-conductive algal cell-mediated hypoxic photosynthesis. a** Chlorophyll fluorescence kinetic curves for PPy/CaCO₃-coated *Chlorella* cells with or without the addition of EY and TEOA. **b** Corresponding spider plots of chlorophyll fluorescence parameters from **a**. **c**–**f** Histograms of chlorophyll fluorescence characteristics (**c**), NADP⁺, NADPH and ATP contents (**d**), Rubisco activity (**e**) and carbohydrate, lipid and protein contents (**f**) for PPy/CaCO₃-coated *Chlorella* cells with or without EY and TEOA. Data are presented as mean values ± SD, error bars indicate standard deviations ($n = 12$ (**c**, $F_v/F_m$, $qN$, $qP$), $n = 4$ (**c**, ETR), $n = 3$ (**d**, **e**, **f**), biologically independent samples). **g**, **h** Time-dependent measurements of hydrogen production for DCMU- (**g**) or DBMIB-treated (**h**) PPy/CaCO₃-coated *Chlorella* cells with or without addition of EY and TEOA (solid lines). The corresponding un-inhibited hydrogen production data are shown by the dotted lines. Data are presented as mean values ± SD, error bars indicate standard deviations ($n = 3$, biologically independent samples). **i** Proposed mechanism of enhanced hydrogen production associated with the extracellular electron-modulated hypoxic photosynthesis of PPy/CaCO₃-coated *Chlorella* algal cells under the addition of extracellular electrons. The extracellular electrons derived from EY and TEOA participate in the photosynthesis pathway (PQ(H₂)→b₆/f→PC→PSI→Fd→hydrogenase), and both photolysis of water and endogenous organic matter contribute to photosynthetic hydrogen generation under hypoxic conditions. All samples are cultivated in sealed vials with sodium ascorbate-containing TAP culture medium and exposed to daylight with an intensity of 65 μE m⁻² s⁻¹. All relevant experiments are performed independently at least three times with similar results. Source data are provided as a Source Data file.

carboxylase (Rubisco) activity (Fig. 4e), decreased carbohydrate content, and elevated lipid and protein concentrations (Fig. 4f). These changes suggested that under the action of extracellular electrons, more endogenous carbohydrate was catabolized through

the NADPH-PQ oxidoreductase, thereby giving rise to increased numbers of electrons for hydrogenase-mediated hydrogen generation. Taken together, the results indicated that both direct and indirect processes of water photolysis were associated with the

successful internalization of the extracellular electrons and contributed to the mechanisms responsible for enhanced hydrogen production.

We also investigated whether the extracellular electrons could be directly utilized as the hydrogenase substrate to increase hydrogen production. To address this, cultures of the PPy/CaCO_3-coated cells with or without EY and TEOA were incubated with 3-(3,4-dichlorophenyl)−1,1-dimethylurea (DCMU) to disrupt electron transfer from $Q_A$ to $Q_B$ in PSII, or with 2,5-dibromo-6-isopropy-3-methyl-1,4-benzoquinone (DBMIB) to inhibit PQ oxidation by the $b_6/f$ complex, impeding electron flow from both PSII and NADPH-PQ oxidoreductase to PSI[40,41]. Addition of DCMU or DBMIB gave rise to a decrease in hydrogen production over 4 d by 75.0% and 98.1%, respectively, with corresponding levels of hydrogen of ~42.0 and 14.0 μmol arising from these two pathways during the 4-days period (Fig. 4g, h), indicating that photolysis of water and endogenous organic matter contributed to hydrogen production (Fig. 4i). As DBMIB did not completely inhibit hydrogen production under the supply of extracellular electrons, the results strongly suggested that the hydrogenase directly received internalized extracellular electrons.

## Photosynthesis-independent hydrogen production

The successful transfer of extracellular electrons under daylight and hypoxic conditions to hydrogenases inside surface-augmented algal cells potentially offers a direct route to hydrogenase-based hydrogen production independent of photosynthesis, providing the possibility of achieving sustainable hydrogen production even in the presence of dead *Chlorella* cells (Fig. 5a). To assess long-term sustainability of the extracellular photosensitizer/sacrificial electron donor system, PPy/CaCO_3-coated algal cells were cultivated in sealed vials with sodium ascorbate-containing TAP culture medium and exposed to daylight with an intensity of $65\,\mu E\,m^{-2}\,s^{-1}$. As shown in Fig. 5b, by supplying extracellular electrons via periodical addition of TEOA at intervals of 6 d, the surface-augmented algal cells were able to maintain hydrogen production for at least 204 d, with a total hydrogen yield of 609 μmol. In comparison, in the absence of EY and TEOA, hydrogen production from the PPy/CaCO_3-coated algal cells was maintained for only 12 d (Fig. 5b). Cell viability was maintained by refreshing with TAP culture medium (as well as ascorbate, EY and TEOA) at 72, 126, and 168 d, as well as re-coating with CaCO_3 at these time points to regenerate an intact cell wall barrier with reduced macromolecular permeability (Supplementary Figs. 20 and 21).

Fluorescence experiments indicated that the maximum photochemical quantum yield of PSII gradually decreased to a $F_v/F_m$ value of *ca.* 0.29 after 120 d (Supplementary Fig. 22). In addition, $F_o$ in the chlorophyll fluorescence curve was gradually decreased over the same period, indicating irreversible damage of PSII that was associated with LHCII dissociation and the impeding of the electron transfer process on the reductant side of PSII (Fig. 5c). This was in agreement with the decreased $F_m$ value, which suggested that the weakened PSII activity arose from conformational changes of D1 protein (Fig. 5c). Moreover, the OJ phase was also gradually destroyed, which indicated that electron transfer in PSII was significantly blocked (Fig. 5c). Nevertheless, the hydrogen production rate only marginally decreased from 3.04 μmol/d for the period of 72 to 126 d, to 2.71 μmol/d between 126 and 168 d, suggesting continued hydrogenase activity in the presence of the extracellular electrons. No OJIP peaks were observed after 204 d, consistent with cell viability experiments, which indicated that nearly all the surface-augmented algal cells were dead (Fig. 5d). Nevertheless, hydrogen production was observed to continue in the presence of the dead cells for a further 8 d with an additional hydrogen yield of 12 μmol (Fig. 5e), after which complete deactivation of the hydrogenases occurred. We attributed the extended hydrogenase activity in the dead cells to their continued enclosure within the PPy/CaCO_3 barrier, which stabilized the ellipsoidal morphology and prevented lysis of the cell contents (Supplementary Figs. 10, 20, and 21).

In conclusion, by combining in situ interfacial polymerization and biomineralization, we demonstrate a cellular bionic system based on the interfacing of algal cells with a conductive PPy/CaCO_3 hybrid shell. Augmentation of the algal cell wall generates a cellular micro-niche that structurally stabilizes the microorganism, facilitates the onset and retention of a localized hypoxic micro-environment and maintains a close-to-neutral environmental pH. Consequently, the coated algal cells switch from the aerobic photosynthesis of oxygen to the hydrogenase-mediated production of hydrogen in air. Our work also demonstrates that photo-generated electrons arising from the coupling of a photosensitizer and sacrificial electron donor in the extracellular environment can be efficiently internalized by the coated algal cell due to facilitated transport through the PPy shell. This in turn boosts the light-dependent and light-independent reactions in the photosynthetic pathway, resulting in enhanced photosynthetic hydrogen production. Significantly, the extracellular photoelectrons can be directly utilized as a hydrogenase substrate to produce hydrogen independent of photosynthesis. Thus, by intermittent replenishing the cell culture with nutrients, EY and TEOA, and regenerating the CaCO_3 exoskeleton, continuous hydrogen generation is achieved for a period of 212 d. Moreover, as the conjugation of hydrogenases to inorganic semiconductors is often compromised by cumbersome purification and protection processes that are required to maintain hydrogenase activity[21], using an algal cell bionic approach circumvents these challenges by directly exploiting functionally active intracellular hydrogenases that operate efficiently due to a stabilized and augmented cellular micro-niche. Thus, integrating artificial biological hydrogen production pathways and natural photosynthesis in surface-augmented algal cells could offer a new paradigm to help solve the current bottleneck associated with short-term biological hydrogen production. More generally, we anticipate that a methodology based on the active interfacing of living cells and polymer/inorganic hybrid materials could provide new bioaugmentation platforms as well as contribute to the development of new cell-based living materials and microbial cell factories with potential applications in sustainable energy production and green bio-manufacturing.

## Methods

### Characterization

Optical and fluorescence microscopy characterization were conducted on a Leica DMI8 manual inverted fluorescence microscope at 10x, 20x, and 40x magnification. X-ray diffraction (XRD) patterns were acquired using a D/MAX-RC thin-film X-ray diffractometer equipped with a nickel filter. The scanning speed was 5°/min and the $2\theta$ range from 10° to 90° at room temperature. Scanning electron microscopy (SEM) images were acquired on a SU8000 instrument with the samples sputter-coated with 10 nm platinum, equipped with an energy-dispersive spectroscope. The pH measurements were conducted with a Seven Compact meter (METTLER TOLEDO, SUI). Transmission electron microscopy (TEM) images were obtained on a JEM-1400, with a filament at 120 kV under bright field mode. Fourier Transform infrared spectroscopy (FTIR) measurements were performed on PerkinElmer spectrometer with a LiTaO_3 detector (Spectrum Two, USA). UV−Vis spectra were measured on a PerkinElmer spectrophotometer (Lambda 750S, USA). Confocal scanning laser microscopy (CSLM) images were obtained on a Leica SP8 confocal laser scanning microscope attached to a Leica DMI 6000 inverted epifluorescence microscope. The dissolved oxygen content was detected using a dissolved oxygen meter (METTLER TOLEDO, F4-Field).

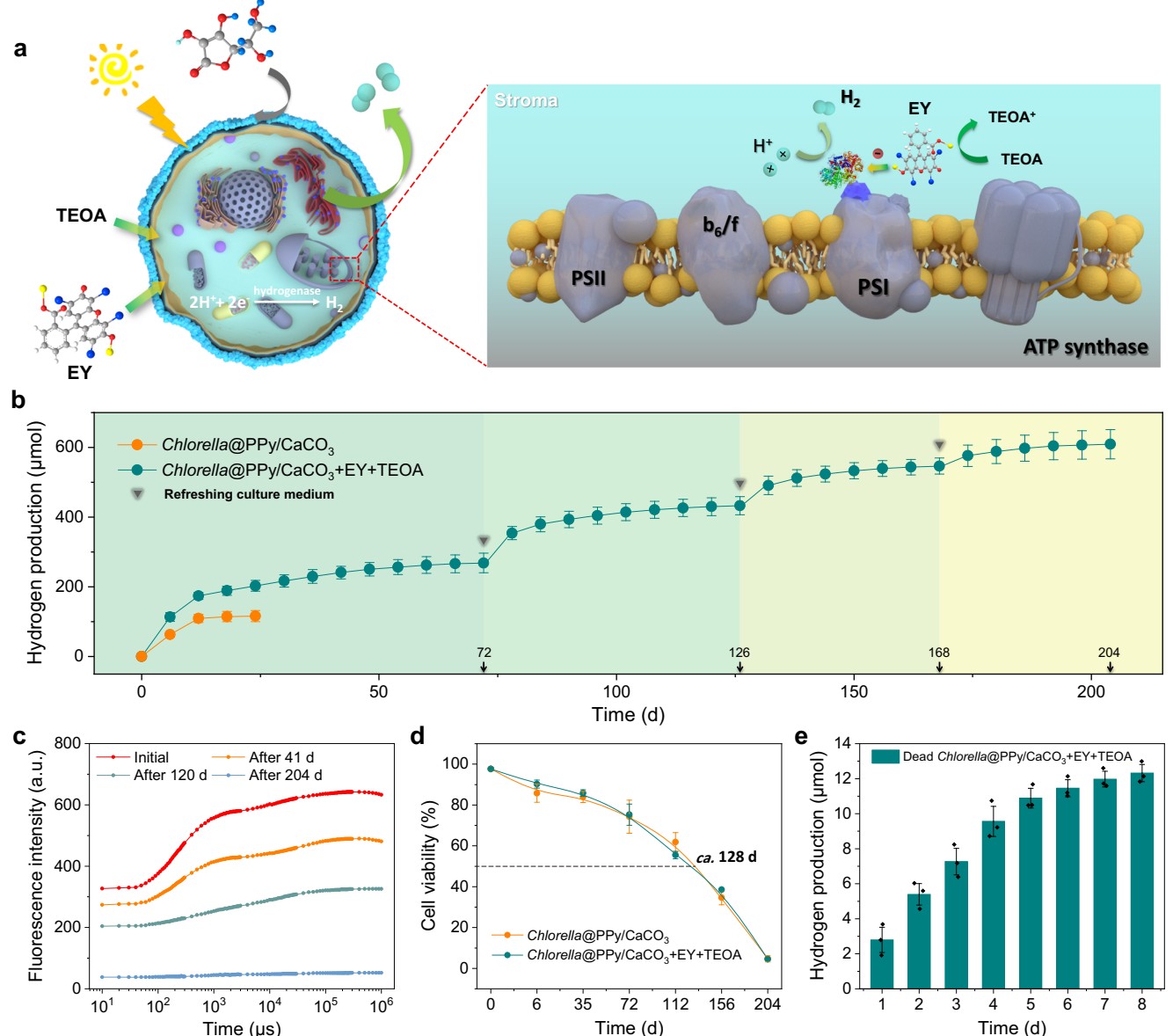

**Fig. 5 | Extracellular electron-mediated photosynthesis-independent extended hydrogen production. a** Schematic illustration of hydrogen production (paired cyan circles) arising from surface-augmented algal cells with the addition of EY and TEOA under hypoxic conditions. The electrons are directly transferred to hydrogenase via molecular diffusion; right panel shows the site of interaction with extracellular electrons. Molecular graphics for EY, TEOA and ascorbate are shown. **b** Time-dependent production of hydrogen for PPy/CaCO$_3$-coated *Chlorella* cells with periodic addition of TEOA at intervals of 6 d from the time point of 12 d. The culture medium containing EY and TEOA is also refreshed (arrows) along with regeneration of the outer CaCO$_3$ shell. The final used concentrations of EY and TEOA at 72 d are 0.23 and 16.8 mg/mL; 126 d, 0.23 and 13.9 mg/mL; 168 d, 0.23 and 11.0 mg/mL; and 204 d, 0.23 and 9.52 mg/mL. Average chlorophyll content obtained within 204 d, 38.3 mg/L. The decrease on the hydrogen production rate is due to normal apoptosis of the *Chlorella* cells. **c** Time-dependent chlorophyll fluorescence kinetics curves for PPy/CaCO$_3$-coated *Chlorella* cells with the addition of EY and TEOA. **d** Time-dependent measurements of cell viability for PPy/CaCO$_3$-coated *Chlorella* cells with or without the addition of EY and TEOA. The half-life of the PPy/CaCO$_3$-coated *Chlorella* cells was ~128 d. **e** Time-dependent measurements of hydrogen production for dead PPy/CaCO$_3$-coated *Chlorella* cells with the addition of EY and TEOA. The final used concentrations of EY and TEOA are 0.24 and 1.65 mg/mL, respectively. All samples are cultivated in sealed vials with sodium ascorbate-containing TAP culture medium and exposed to the daylight with an intensity of 65 μE m$^{-2}$ s$^{-1}$. Data in **b**, **d**, and **e** are presented as mean values ± SD, error bars indicate standard deviations ($n$ = 3, biologically independent samples). All relevant experiments are performed independently at least three times with similar results. Source data are provided as a Source Data file.

## Cell cultures

*Chlorella pyrenoidosa* cells were cultured in TAP medium composed of $2 \times 10^{-2}$ M Tris, $7 \times 10^{-3}$ M NH$_4$Cl, $8.3 \times 10^{-4}$ M MgSO$_4$·7H$_2$O, $4.5 \times 10^{-4}$ M CaCl$_2$·7H$_2$O, $1.65 \times 10^{-3}$ M K$_2$HPO$_4$, $1.05 \times 10^{-3}$ M KH$_2$PO$_4$, 1 mL/L Hunter's trace elements and 1 mL/L glacial acetic acid. The medium was sterilized beforehand, and the pH adjusted to 7.2. *Chlorella* cells were cultured in 500 mL of TAP medium at 25 °C with alternative daytime (12 h, 40 μE m$^{-2}$ s$^{-1}$) and night (12 h) exposures in a light incubator. The number of cells was determined by optical density measurements at

750 nm (OD$_{750}$) using UV–Visible spectroscopy. The chlorophyll concentration was determined spectrophotometrically using 95 % (v/v) ethanol.

## Coating of algal cells

The *Chlorella* cells were thoroughly washed with NaCl aqueous solution (0.01 M) and DI water for three times by a centrifugation method ($3775 \times g$, 3 min). For the initial coating of PPy, 200 μL of the cell suspension were added to 5 mL of an aqueous solution of iron (III)

chloride nonahydrate (1 mg/mL, 6.2 mM, pH = 2.4) to give a suspension (OD$_{750}$ = 3.0), and left stirring at 400 rpm for 30 min, followed by centrifugation (3775 × $g$, 3 min) to remove unbound Fe (III). The Fe(III)-treated algal cells were re-suspended in 5 mL of DI water, and 10 μL of pyrrole was then added into the mixture. The mixture was stirred (400 rpm) at room temperature for 6 h, during which in situ Fe(III)-mediated oxidative polymerization of pyrrole occurred on the cell wall. The samples were washed with DI water to remove residual pyrrole monomers and the reduced co-product (Fe(II)). Fe(III)-doped PPy-coated *Chlorella* cells were then coated with CaCO$_3$, by immersion in 1 mL of 0.3 M aqueous CaCl$_2$ solution, followed by incubation with stirring (400 rpm) at room temperature for 20 min. Then, 1 mL of 0.3 M aqueous Na$_2$CO$_3$ solution were added, and the mixture stirred at 400 rpm for 5 min. The PPy/CaCO$_3$-coated cells were obtained after thorough washing with DI water for three times to remove any residual reactants in the solution.

## Cyclic voltammetry characterization
The native and PPy-coated *Chlorella* cells were washed using DI water and centrifugation (3775 × $g$, 3 min), and then immersed in nafion solution (0.5%, v/v). Then, one drop of the cell suspension was added onto a carbon cloth substrate and dried overnight at room temperature. For fabrication of a three-electrode system, the working electrode was the cell-attached carbon cloth substrate, a platinum wire was the counter electrode, and Ag/AgCl was used as reference electrode. PBS (pH = 7.0, 0.1 M) containing 100 mM of NaCl was chosen as the electrolyte. The samples were then conducted on the electrochemical workstation (Chenhua CHI604E, Shanghai, China) at the scanning rate of 200 mV/s.

## Determination of chlorophyll fluorescence kinetics
One milliliter of native *Chlorella* cells, PPy-coated cells and PPy/CaCO$_3$-coated algal cells were added into sample tubes and kept in the dark for 20 min to ensure that the PSII reaction centers were completely open and the photosynthetic electron transfer chain thoroughly oxidized. The samples were then determined by a chlorophyll fluorometer (Yaxin, 1161G, Beijing, China) to acquire chlorophyll fluorescence kinetic curves. The maximum quantum yield of PSII ($F_v/F_m$) that was indicative of the photosynthetic activity of the algal cells was directly acquired, and other photosynthetic parameters were calculated from the chlorophyll fluorescence curves (see Supplementary Table 2 and Supplementary Note 3 for details).

## Measurement of algal conductive and mechanical properties
*Chlorella* cells were thoroughly washed using DI water. The aqueous dispersions were mounted and air-dried onto a silica/ITO wafer. Measurements of surface current and Young's modulus were undertaken using an atomic force microscope (AFM) equipped with a 3D manipulation stage.

## Cell viability tests
The stock solution of fluorescein diacetate (FDA) was prepared by dissolving FDA in acetone at a concentration of 5 mg/mL. To label the viable *Chlorella* cells, 5 μL of FDA solution was added to 1 mL of aqueous *Chlorella* cell dispersions. After incubation in dark at room temperature, the labeled cells were washed with DI water for three times and imaged in a fluorescence microscope.

## Photosynthetic hydrogen production
A constant volume of the native, PPy-coated or PPy/CaCO$_3$-coated *Chlorella* cells was transferred into a 100 mL sample bottle (10 mL TAP culture medium and 90 mL head air space). The final concentration of chlorophyll was ~13 μg/mL. Then, 200 mg of sodium ascorbate was added into the culture medium and allowed to dissolve by shaking the flask by hand. The samples were then exposed to a light intensity of

65 μE m$^{-2}$ s$^{-1}$ and shaken throughout at a rate of 150 rpm. Hydrogen evolution driven by extracellular electrons was undertaken using Eosin Y disodium salt (EY) and triethanolamine (TEOA) as the photosensitizer and electron donor, respectively. Bulk solutions of EY and TEOA were prepared at concentrations of 1.5 and 200 mg/mL, respectively. In both cases, the pH was adjusted to 7.2 and dissolved oxygen thoroughly removed by nitrogen bubbling for 20 min prior to use. To generate artificial extracellular electrons around the algal cells, 2 mL of the EY solution and 100 μL of the TEOA solution were introduced into the hydrogen production system, with final used concentrations of EY and TEOA of 0.24 and 1.65 mg/mL, respectively. Time-dependent evolution of hydrogen gas in the samples were monitored using a Hydrogen Detector (AP-B-H$_2$-F; capacity, 5000 ppm; resolution ratio, 1 ppm). The hydrogen production rate was calculated according to the total content of chlorophyll associated with the algal cells in the culture medium.

## Immunoblot analysis of hydrogenase
The *Chlorella* cells were washed using DI water and re-suspended in 5× protein lysis buffer (0.25 M TrisHCl, pH = 8.0; 25% glycerol; 0.25 mg/mL bromophenol blue; 7.5% SDS; 12.5% 2-mercaptoethanol). The samples were heated for 10 min at 100 °C, followed by centrifugation at 10,000 × $g$ for 1 min. The supernatant was then separated by SDS-PAGE (sodium dodecyl sulfate polyacrylamide gel electrophoresis), and the gels were blotted onto polyvinylidene difluoride (PVDF) membrane for the detection of hydrogenase. Anti-hydrogenase A (Agrisera) and goat anti-rabbit IgG (H&L)-HRP conjugate (Agrisera) were used as the primary antibody and secondary antibody, respectively.

## Measurement of ATP, NADP$^+$, NADPH, protein content, and Rubisco activity
Typically, the *Chlorella* cells were washed using DI water and diluted with PBS solution (pH = 7.4). The cells were lysed by repeated alternate freezing and thawing cycles, and the mixture was centrifuged (3775 × $g$, 3 min) to obtain the supernatant. The concentration of ATP, NADP$^+$, NADPH, and the activity of Rubisco were then determined spectrophotometrically with the Elisa Kit (Detect Technical Institute, Shanghai, China). The protein content was tested using a BCA Protein Assay Kit.

## Measurement of intracellular carbohydrate content
*Chlorella* cells were washed using DI water and collected by centrifugation (3775 × $g$, 3 min). The cells were then placed into a freezer drier for 24 h. For the extraction of carbohydrate, the cell powder was immersed in 10 mL of 80% ethanol and stirred continuously at 80 °C for 30 min, followed by centrifugation (5285 × $g$, 5 min) to acquire the supernatant. The powder was repeatedly extracted using the above procedures. Then, 1 mL of extraction solution was added to 5 mL of anthrone solution (in 80% sulfuric acid, 1 mg/mL). The mixture was kept at 100 °C for 10 min and the carbohydrate content determined spectrophotometrically at an absorbance of 625 nm (OD$_{625}$).

## Measurement of intracellular lipid content
*Chlorella* cells were washed using DI water and collected by centrifugation (3775 × $g$, 3 min), and then freeze-dried for 24 h to produce the cell powder. The dried cells were weighed ($M_1$) and immersed in diethyl ether for 16 h and then placed into a Soxhlet extractor for extraction of lipid. The powder was dried and weighed ($M_2$), and the lipid content ($N$) was calculated by Eq. (1):

$$N (\%) = \frac{M_1 - M_2}{M_1} * 100 \tag{1}$$

## Measurements of the chlorophyll content

A total of 300 μL of *Chlorella* cells containing culture medium was transferred into a centrifuge tube. After washing three times using DI water, the cells were transferred into 300 μL of ethanol. The mixture was thoroughly homogenized, and then left at 4 °C for at least 12 h. The solution was then centrifuged (3775 × *g*, 3 min) and the supernatant withdrawn to determine the chlorophyll content by spectroscopic measurements at absorbance values of 649 nm ($A_{649}$) and 665 nm ($A_{665}$). The total concentration of chlorophyll (*C*, μg/mL) was calculated using the Eq. (2):

$$C = 6.1 * A_{665} + 20.4 * A_{649} \qquad (2)$$

## Reporting summary

Further information on research design is available in the Nature Portfolio Reporting Summary linked to this article.

## Data availability

Data supporting the findings of this work are available within the paper and its Supplementary Information files. A reporting summary for this Article is available as a Supplementary Information file. Source data are provided with this paper.

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

## Acknowledgements

We thank NSFC (22171058 and 21871069) and the Fundamental Research Funds for the Central Universities (HIT.OCEF.2021027) for financial support.

## Author contributions

Z.X. and X.H. conceived the experiments; Z.X., J.Q., S.W. performed the experiments; Z.X., J.Q., S.W., X.L., M.L., X.H., and S.M undertook data analysis; Z.X., X.H., and S.M. wrote the manuscript.

## Competing interests

The authors declare no competing interests.
