## [Peer Review File · Nature Communications]

Algal cell bionics as a step towards photosynthesis-independent hydrogen productionReviewers' Comments:

Reviewer #1:

Remarks to the Author:

This paper outlines a novel encapsulation of intact algal cells that creates a micro-anaerobic compartment. The novelty of this work involves combining two well-known coating chemistries with studying a photosynthetic green alga. This multi-layer coating provides an electrochemically conductive surface composed of FeCl₃ and pyrrole and a relatively inert yet mesoporous calcium carbonate exterior supportive layer. The simplicity, ease of control in fabrication, and the use of largely earth-abundant materials make this approach very attractive to many other in vivo and in vitro biological processes. Although the calcium carbonate coating has already been shown to be electrochemically porous when entrapping isolated photosystems, this work is the second work that extends this standard pharmacological procedure to photosynthetic systems. It now applies to a larger, intact cellular system, further demonstrating the high biocompatibility of CaCO₃ microspheres.

Although this paper offers a wealth of physical and rheological characterizations that help explain much of the biological observations. It is surprisingly free of some fundamental biochemical characterizations. For example, there is no immunoblotting or proteomics to look for the expression of the FeFe hydrogenase induced under these conditions, nor is there any work to look at other changes in the photosynthetic complexes that have been previously shown to undergo remodeling during hypoxic exposure in *Chlorella*. In addition, although this work is excellent, I feel it should recognize prior advances in highly related subjects in many places. For example, both coating chemistries are well established, yet this work does not fully divulge that information. Some of these references are listed below, yet it is not comprehensive.

Specific comments:

- 1) Overall, the paper is well constructed; however, in light of the large number of relatively detailed and specialized photochemical characterizations (Fig. 4 a-c and Fig. 5c.), I would request that they provide a bit more information and explanation on how these results are to be interpreted.
- 2) The reader should understand that electrons are being derived from both PSII and direct substrate donation- this is clear from the DCMU and DBMIB effects, yet on Page 6, line 188, they indicate that the electrons originate mainly from PSII. This, plus some of the references below, need to be explained better.
- 3) Related to the two sources of electrons, the cartoon in Figure 4i has substrate-level electrons coming into a Blue Ball? What is this supposed to signify? It is clearly between PSII and b6/f, but what is the blue ball? The PQH₂ pool?
- 4) In Fig. 4g and h, they should add the un-inhibited data from Fig. 3d for direct comparison.
- 5) Overall, I find the heavy use of orange and red in the graphs challenging to differentiate, especially in the Kautsky curves (Fig. 4a and Fig. 5c)
- 6) The cell viability data in Fig. 5d should be graphed in an XY plot time on the x-axis, not as a bar graph. From this XY graph, it would allow a half-life to be inferred and/or possibly mathematically fit.
- 7) It is unclear why the hydrogen yield continues to increase as a function of time with the dead cells.
- 8) Many images are too small to be informative- The insert in Fig. 1g; the inserts in Fig. 3b and c.
- 9) What is the spectral response of eosin yellow? It absorbs around 520 nm and should possibly allow the precise contribution to be teased apart.
- 10) What are the spectral properties of the light used? Only an intensify in flux is provided. This is important in interpreting the input of the EY results. The use of LEDs would help provide better spectral control.
- 11) The final concentrations of both EY and TEOA should also be provided for these experiments. It is unclear if this can be calculated from the data provided.
- 12)

However, even in light of these comments, there are multiple novel features of this work that warrants its publication:

- 1) A simple system to create a micro-anaerobic environment to support hydrogenase expression and activity without the need to induce nutrient deprivation
- 2) Integration of a dual cellular coating that allows electrochemical connectivity to a sacrificial electron donor and photosensitizer
- 3) A stable system that preserves viability for many months with a half-life of ~200 days)

Missing relevant citations:

- 10.1007/s10529-011-0584-x
- <https://doi.org/10.1016/j.algal.2020.101827>
- <https://doi.org/10.1016/j.biortech.2019.121762>
- <https://doi.org/10.1016/j.mtmbio.2021.100122>
- <http://dx.doi.org/10.1038/nnano.2009.315>
- 10.1007/s10854-018-0510-2

Reviewer #2:

Remarks to the Author:

In this manuscript, the authors report the Algal cell bionics as a step towards photosynthesis-independent hydrogen production. In view of the inherent vulnerability of individual cells, it is of great significance to develop integrated techniques with good biocompatibility and superior environmental tolerance.

The manuscript is well written, and most of its content is well described. The coated algal cells switch from the aerobic photosynthesis of oxygen to the hydrogenase-mediated production of hydrogen in air. This is a new attempt in this article. On the other hand, I found that some of the background information in the thesis is somewhat general, while some important points are not fully described. Thus, I recommend its publication for a Major revisions after the authors address the following questions:

1. When exogenous materials are introduced into cells, their toxicities will increase significantly, resulting in cell function damage and even cell death. Can the authors clarify the effects of the PPy/CaCO₃-coating methodology on the algal cells, to determine if there were any side effects related to toxicity?
2. Although the H₂ production increased in a short period of time, the rate of this system started to decrease due to destructed aggregation structures resulting from cell proliferation. how surface-augmented algal cell changes during the course of long working hours is unclear. The authors should further discuss them.
3. In Figure 5, EY and TEOA in the cell is not easily fixed because of the fluidity of the cytoplasm, making it difficult to bind the hydrogenase. Can the authors explain the proposed electron transfer mechanism more detailly?

Reviewer #3:

Remarks to the Author:

This study reports a nanobiohybrid cell system based on the interfacing of algal cells with a conductive PPy/CaCO₃ hybrid shell. The hybrid coating structurally stabilizes the microorganism, facilitates the onset and retention of a localized hypoxic micro-environment. Consequently, the coated algal cells

switch from the aerobic photosynthesis of oxygen to the hydrogenase-mediated production of hydrogen in air. By addition of a photosensitizer and sacrificial electron donor in the extracellular environment, it can boost the light-dependent and light-independent reactions in the photosynthetic pathway, resulting in enhanced photosynthetic hydrogen production. Overall, it is an interesting work. I suggest some issues for the manuscript should be revised before being considered for publication.

-Fig 1e is not very convincing. Also the absorption values are too high, you are probably saturating the detector. You need to reduce the concentration.

-In Fig 1g, it is not clear which section of the cell is imaged. Force measurements need to be done on a few different areas and average them out.

-“nutrients in the external media remained accessible to the live *Chlorella* cells after coating of the cell wall”- Can you specify what are the nutrients and molecular sizes of these nutrient molecules?

-In Fig S14, the coating improves thermal and acid stability. This is good results, but lacks explanation. The coating is just on the cell surface, how can it help maintain cell activities, which are happening inside the cells. In addition, wouldn't acid (pH 2) dissolves CaCO₃ coating? Even if not, the porosity would allow acid to diffuse into the cells.

-Beside the scheme in Fig 2A, the mechanism of O₂ consumption by Fe(III)-doped PPy is not clearly described and proved.

-Fig 2d, why pH drop? It is not explained.

-In Fig 3d, the augmentation of photoelectron transfer should be also demonstrated using uncoated algae as a control.

-The step-like pattern in H₂ production is probably an indication that the algae are not very active after a few days following new EY and TEOA and CaCO₃ addition. Can you reduce the interval of EY and TEOA and CaCO₃ renewal to optimize the H₂ production?

-Also it seems the coating is not very stable after a few days of operation. Can you provide an explanation and possible further improvements?

Reviewer #1 (Remarks to the Author):

This paper outlines a novel encapsulation of intact algal cells that creates a micro-anaerobic compartment. The novelty of this work involves combining two well-known coating chemistries with studying a photosynthetic green alga. This multi-layer coating provides an electrochemically conductive surface composed of FeCl₃ and pyrrole and a relatively inert yet mesoporous calcium carbonate exterior supportive layer. The simplicity, ease of control in fabrication, and the use of largely earth-abundant materials make this approach very attractive to many other in vivo and in vitro biological processes. Although the calcium carbonate coating has already been shown to be electrochemically porous when entrapping isolated photosystems, this work is the second work that extends this standard pharmacological procedure to photosynthetic systems. It now applies to a larger, intact cellular system, further demonstrating the high biocompatibility of CaCO₃ microspheres.

Response: Many thanks for the reviewer's very positive comments on our work.

Although this paper offers a wealth of physical and rheological characterizations that help explain much of the biological observations. It is surprisingly free of some fundamental biochemical characterizations. For example, there is no immunoblotting or proteomics to look for the expression of the FeFe hydrogenase induced under these conditions, nor is there any work to look at other changes in the photosynthetic complexes that have been previously shown to undergo remodeling during hypoxic exposure in *Chlorella*.

Response: Thank you for the helpful comments. As suggested, immunoblot (Western Blot) analysis has now been conducted to verify the expression of algal [FeFe]-hydrogenase at different time points within the hypoxic period. The new data has been added as **Supplementary Fig. 16** in SI, which indicates the gradual expression of hydrogenase in the anaerobic environment induced by the Fe(III)-mediated oxidation of ascorbate.

Supplementary Figure 16. HYDA Western Blot analysis of native *Chlorella* cells in normal TAP culture medium and PPy/CaCO₃-coated *Chlorella* cells in ascorbate-containing TAP culture medium at different time points. Lane 1: marker; Lane 2: native *Chlorella* cells in normal TAP; Lane 3, 4, 5, 6: PPy/CaCO₃-coated *Chlorella* cells in ascorbate-containing TAP culture medium at the time points of 1, 3, 5, 7 d. The time-dependent deepening of band color indicates the gradual expression and activation of hydrogenase in the anaerobic environment induced by the Fe(III)-mediated oxidation of ascorbate.

The relevant description of experimental procedures has been added into Supporting Information-Methods on Page 4:

“Immunoblot analysis of hydrogenase. The *Chlorella* cells were washed using DI water and

resuspended in 5x protein lysis buffer (0.25 M TrisHCl, pH = 8.0; 25 % glycerol; 0.25 mg/mL bromophenol blue; 7.5 % SDS; 12.5 % 2-mercaptoethanol). The samples were heated for 10 min at 100 °C, followed by centrifugation at 10000 g for 1min. The supernatant was then separated by SDS-PAGE (sodium dodecyl sulfate polyacrylamide gel electrophoresis), and the gels were blotted onto polyvinylidene difluoride (PVDF) membrane for the detection of hydrogenase. Anti-hydrogenase A (Agriser) and goat anti-rabbit IgG (H&L)-HRP conjugate (Agriser) were used as the primary antibody and secondary antibody, respectively.”

In addition, although this work is excellent, I feel it should recognize prior advances in highly related subjects in many places. For example, both coating chemistries are well established, yet this work does not fully divulge that information. Some of these references are listed below, yet it is not comprehensive.

Response: Many thanks for the suggestions. We have now included more references recognizing prior advances in the Introduction section; 8 new references are cited along with corresponding descriptions in the main text on Page 2. New references are cited as Refs 15-17, 32-34 and 37-38. The corresponding text reads:

“Consequently, strategies based on anaerobic fermentation, respiration enhancement, nutrient deprivation and gene engineering of oxygen-tolerant hydrogenases have been developed to eliminate the negative impact of oxygen production^{10, 11, 12, 13, 14, 15, 16, 17}.”

“Recently, mesoporous microspheres,³² platinum nanoclusters³³ and polypyrrole³⁴ have been employed to fabricate biohybrids capable of cell-free hydrogen production, tunable electrical conductivity or self-enhancing photoactivity. In addition, construction of bionic functional layers around living cells with ultrathin shells of a coacervate phase,³⁵ semiconductors and carbon nanotubes,³⁶ polypyrrole³⁷ or calcium carbonate³⁸ has proved to be an effective strategy for enhancing natural cellular properties and augmenting living organisms with non-natural functions.”

The added new references are:

Ref15. Li L, Zhang L, Gong F, Liu J. Transcriptomic analysis of hydrogen photoproduction in *Chlorella pyrenoidosa* under nitrogen deprivation. *Algal Res.* 47, 101827 (2020).

Ref16. Liu J-Z, et al. Exogenic glucose as an electron donor for algal hydrogenases to promote hydrogen photoproduction by *Chlorella pyrenoidosa*. *Bioresour. Technol.* 289, 121762 (2019).

Ref17. Wang H, Fan X, Zhang Y, Yang D, Guo R. Sustained photo-hydrogen production by *Chlorella pyrenoidosa* without sulfur depletion. *Biotechnol. Lett.* 33, 1345-1350 (2011).

Ref32. Teodor AH, Thal LB, Vijayakumar S, Chan M, Little G, Bruce BD. Photosystem I integrated into mesoporous microspheres has enhanced stability and photoactivity in biohybrid solar cells. *Mater. Today Bio* 11, 100122 (2021).

Ref33. Iwuchukwu IJ, Vaughn M, Myers N, O'Neill H, Frymier P, Bruce BD. Self-organized photosynthetic nanoparticle for cell-free hydrogen production. *Nat. Nanotechnol.* 5, 73-79 (2010).

Ref34. Vellguth N, Shamsuyeva M, Kroll S, Renz F, Endres H-J. Electrical conductivity in

biocomposites via polypyrrole coating. *J. Mater. Sci.: Mater. Electron.* 30, 2373-2381 (2019).
Ref37. Song RB, et al. Living and conducting: coating individual bacterial cells with in situ formed polypyrrole. *Angew. Chem. Int. Ed.* 129, 10652-10656 (2017).
Ref38. Fakhruddin RF, Minullina RT. Hybrid cellular-inorganic core-shell microparticles: encapsulation of individual living cells in calcium carbonate microshells. *Langmuir* 25, 6617-6621 (2009).

Specific comments:

1) Overall, the paper is well constructed; however, in light of the large number of relatively detailed and specialized photochemical characterizations (Fig. 4 a-c and Fig. 5c.), I would request that they provide a bit more information and explanation on how these results are to be interpreted.

Response: Many thanks for the reviewer's suggestions. More information and explanations focusing on **Fig. 4a-c** and **Fig. 5c** are now provided. In general, the calculation of the photosynthetic parameters is based on the formula in the section "Supporting Information-Methods-Determination of chlorophyll fluorescence kinetics", and their definitions and explanations are summarized in **Supplementary Table 2**. The interpretations on the **Fig. 4a-c** have been revised in the main text on Page 9 (yellow highlighted sections), which now read as follows:

"To confirm the role of extracellular electrons in augmented hypoxic photosynthesis, we measured chlorophyll fluorescence transient curves and fluorescence kinetic parameters to evaluate the absorption and capture of light energy by PSII, as well as the subsequent photosynthetic electron transfer processes for PPy/CaCO₃-coated cells in the presence or absence of EY and TEOA (**Fig. 4a**). For the PPy/CaCO₃-coated cells under the addition of EY and TEOA to the extracellular medium, F_m in the chlorophyll fluorescence curve was increased, which suggested that the activity of D1 protein in PSII was enhanced, and thus contributed to the higher efficiency of PSII electron acceptors (**Fig. 4a**). Moreover, an elevated quantum yield for electron transport (ϕ_{E0}) was observed, indicating that the captured photoenergy from PSII was more efficiently utilized for subsequent transfer (**Fig. 4b**). Combining this with the enhanced S_m value (energy required to completely reduce Q_A), it was suggested that the PQ pool was enlarged, thus indicating more electrons were transferred throughout the photosynthetic chain (**Fig. 4b**). In addition, the density of the PSII reaction centers was also increased, as indicated by the improved number of PSII reducing centres per CS_m (RC/CS_m) (**Fig. 4b** and **Supplementary Table 2**). In terms of the specific energy flux in the photosystems, for the PPy/CaCO₃-coated cells in the presence of EY and TEOA, the adsorbed energy (ABS) per excited cross-section (CS_m) (ABS/CS_m) and trapped energy (TR) per CS_m (TR_0/CS_m) were enhanced (**Fig. 4b**, and **Supplementary Table 2**), which suggested that more photoenergy was absorbed by chlorophyll and then utilized for the reduction of Q_A . Furthermore, the energy used for both electron transfer (ET_0) per CS_m (ET_0/CS_m) and reducing end electron acceptors (RE_0) per CS_m (RE_0/CS_m) were increased (**Fig. 4b**, and **Supplementary Table 2**), indicating that the reoxidation of the reduced Q_A along with the electron transport was improved and more

electrons reached the end of electron transferring chain. These observations revealed that the extracellular electrons participated in the photosynthesis pathway and improved all the efficiencies of photoenergy absorption, capture, and transfer in the photosynthetic chain of *Chlorella* cells. Consequently, the performance indices based on absorption (PI_{abs}), cross-section (PI_{cs}) and energy conversion (PI_{total}), and F_v/F_m values were all considerably improved (**Fig. 4b** and **Supplementary Table 2**), indicating successful internalization of extracellular electrons by the PPy/CaCO₃-coated *Chlorella* cells and an increase in photoactivity.”

The interpretations on **Fig. 5c** have been revised in the main text on Page 12 yellow highlight parts, which now read as follows:

“Fluorescence experiments indicated that the maximum photochemical quantum yield of PSII gradually decreased to a F_v/F_m value of *ca.* 0.29 after 120 days (**Supplementary Fig. 21**). In addition, F_o in the chlorophyll fluorescence curve was gradually decreased over the same period, indicating irreversible damage of PSII that was associated with LHCII dissociation and the impeding of the electron transfer process on the reductant side of PSII (**Fig. 5c**). This was in agreement with the decreased F_m value, which suggested that the weakened PSII activity arose from conformational changes of D1 protein (**Fig. 5c**). Moreover, the OJ phase was also gradually destroyed, which indicated that electron transfer in PSII was significantly blocked (**Fig. 5c**).”

2) The reader should understand that electrons are being derived from both PSII and direct substrate donation- this is clear from the DCMU and DBMIB effects, yet on Page 6, line 188, they indicate that the electrons originate mainly from PSII. This, plus some of the references below, need to be explained better.

Response: The sentence on Page 6 (line 188) refers to general conditions rather than later experiments with DCMU and DBMIB. We do qualify this by stating that the photoelectrons are **mainly** derived from PSII, and have therefore not revised this text. With regard to the contribution from external substrates we have made a clear statement to this effect on Page 10 where we write “*Addition of DCMU or DBMIB gave rise to a decrease in hydrogen production over 4 days by 75.0 and 98.1 %, respectively, with corresponding levels of hydrogen of approximately 42.0 and 14.0 μ mol arising from these two pathways during the 4-day period (Fig. 4g,h), indicating that photolysis of water and endogenous organic matter contributed to hydrogen production (Fig. 4i).*”

This is also consistent with other reported work showing that electrons for hydrogen production are from both PSII and endogenous organic matter. Two new references (Ref 40,41) have now been included on Page 10.

3) Related to the two sources of electrons, the cartoon in Figure 4i has substrate-level electrons coming into a Blue Ball? What is this supposed to signify? It is clearly between PSII and b_6/f , but what is the blue ball? The PQH₂ pool?

Response: The blue ball in **Fig.4i** signifies the PQ(H₂) pool. We have added the label “PQ(H₂)” to the graphic in revised **Fig.4i**.

4) In Fig. 4g and h, they should add the un-inhibited data from Fig. 3d for direct comparison.

Response: We have added the un-inhibited data. The revised **Fig. 4g** and **Fig. 4h** are as follows:

Revised Figure 4g-h. Time-dependent measurements of hydrogen production for DCMU- (**g**) or DBMIB-treated (**h**) PPy/CaCO₃-coated *Chlorella* cells with or without addition of EY and TEOA (solid lines). The corresponding un-inhibited hydrogen production data are shown by the dotted lines. Data are presented as mean values ± SD, error bars indicate standard deviations (n = 3).

5) Overall, I find the heavy use of orange and red in the graphs challenging to differentiate, especially in the Kautsky curves (Fig. 4a and Fig. 5c)

Response: The color schemes in old **Fig. 3d** (now **Fig. 3f**), **Fig. 4a-h**, **Fig. 5b**, **Fig. 5d-e** and **Supplementary Fig. 22** have been refined to make the graphs more easily differentiated.

6) The cell viability data in Fig. 5d should be graphed in an XY plot time on the x-axis, not as a bar graph. From this XY graph, it would allow a half-life to be inferred and/or possibly mathematically fit.

Response: Many thanks for the suggestion. **Fig. 5d** has been revised as an XY plot, shown below. The half-life of the PPy/CaCO₃-coated *Chlorella* cells is calculated to be about 128 days.

Revised Figure 5d. Time-dependent measurements of cell viability for PPy/CaCO₃-coated *Chlorella* cells with or without the addition of EY and TEOA. The half-life of the PPy/CaCO₃-coated *Chlorella* cells was approximately 128 days.

7) It is unclear why the hydrogen yield continues to increase as a function of time with the dead cells.

Response: On Page 12 we write: “We attributed the extended hydrogenase activity in the dead cells to their continued enclosure within the PPy/CaCO₃ barrier, which stabilized the ellipsoidal morphology and prevented lysis of the cell contents (**Supplementary Figs. 10, 20 and 21**).”

8) Many images are too small to be informative- The insert in Fig. 1g; the inserts in Fig. 3b and c.

Response: Many thanks for pointing this out. **Fig. 1g** associated with the force measurement has been changed into a more accurate bar graph. The inserts in original **Fig. 3b,c** have been enlarged as sub-graphs in the revised **Fig. 3**, as shown below:

Revised Figure 3. Enhanced extracellular photoelectron transport and hydrogen production in surface-conductive algal cells. **a**, Schematic illustration of the PPy shell-mediated capture and translocation of artificially generated extracellular electrons for enhanced hydrogen production (paired cyan circles) in surface-coated algal cells under air; molecular graphics for EY, TEOA and ascorbate are shown. **b,c**, Histograms (**b**) of measured surface current for native *Chlorella* cells with the corresponding value-distribution image (**c**). **d,e**, Histograms (**d**) of measured surface current for PPy-coated *Chlorella* cells with the corresponding value-distribution image (**e**). The single peak fitting was performed using a Gaussian curve in the histograms. **f**, Time-dependent measurements of hydrogen production for PPy/CaCO₃-coated *Chlorella* cells with or without EY and TEOA. EY and TEOA were added at time point of 1 day. Samples were cultivated in seal vials with sodium ascorbate-containing TAP culture medium and exposed to daylight with an intensity of 65 $\mu\text{E m}^{-2} \text{s}^{-1}$. Data are presented as mean values \pm SD, error bars indicate standard deviations ($n = 3$).

9) What is the spectral response of eosin yellow? It absorbs around 520 nm and should possibly allow the precise contribution to be teased apart.

Response: Many thanks for the enquiry. The spectral property of the used eosin Y (EY) is shown below with the absorbance at around 520 nm, and emission at around 540 nm under excitation at 520 nm. In this study, EY as the photosensitizer receives the photoenergy from the light with the wavelength at around 520 nm, coupled with the TEOA as sacrificial agent to generate extracellular electrons around PPy/CaCO₃-coated *Chlorella* cells for the enhanced photosynthetic hydrogen production. The emission of EY itself does not obviously influence the algal photosynthesis because the chloroplast shows minimal absorbance around the wavelength of 540 nm.

Response Figure 1. Plot of UV-Vis absorbance spectrum and emission spectrum of EY.

10) What are the spectral properties of the light used? Only an intensity in flux is provided. This is important in interpreting the input of the EY results. The use of LEDs would help provide better spectral control.

Response: Given the normal photosynthesis of the alga cell, the constructed bionic system aims to work under natural daylight, and a white fluorescent lamp (104 W) is specially used to simulate the sunlight within the visible light range. The spectral property of the used white fluorescent lamp is shown below, which has been added into SI on Page 15 as **Supplementary Fig. 17**.

Supplementary Figure 17. The spectrum of light source of used white fluorescent lamp (104 W).

Based on the above spectrum, the light source can provide enough photoenergy for the excitation of EY to generate extracellular electrons under the coupling of TEOA. In addition, we agree that LED light could provide light power with the specific wavelength for both photosynthesis of algal cell and excitation of EY, which means better spectral control of the light source. Many thanks for the insightful suggestion, which we will follow up in future work.

11) The final concentrations of both EY and TEOA should also be provided for these experiments. It is unclear if this can be calculated from the data provided.

Response: Many thanks for pointing it out. The final concentrations of both EY and TEOA have been provided both in Supporting Information-Methods-Photosynthetic hydrogen production (Page 3), and in the caption of **Figure 5b** and **5e** on Page 13.

12) However, even in light of these comments, there are multiple novel features of this work that warrants its publication:

- 1) A simple system to create a micro-anaerobic environment to support hydrogenase expression and activity without the need to induce nutrient deprivation
- 2) Integration of a dual cellular coating that allows electrochemical connectivity to a sacrificial electron donor and photosensitizer
- 3) A stable system that preserves viability for many months with a half-life of ~200 days)

Response: We appreciate the reviewer's positive and valuable comments on the scientific merit of our work.

13) Missing relevant citations:

- [10.1007/s10529-011-0584-x](https://doi.org/10.1007/s10529-011-0584-x)
- <https://doi.org/10.1016/j.algal.2020.101827>
- <https://doi.org/10.1016/j.biortech.2019.121762>
- <https://doi.org/10.1016/j.mtbio.2021.100122>
- <http://dx.doi.org/10.1038/nnano.2009.315>
- [10.1007/s10854-018-0510-2](https://doi.org/10.1007/s10854-018-0510-2)

Response: Many thanks for pointing these out. The missing relevant citations have been added in the main text as Ref 15-17 and 32-34.

Reviewer #2 (Remarks to the Author):

In this manuscript, the authors report the Algal cell bionics as a step towards photosynthesis-independent hydrogen production. In view of the inherent vulnerability of individual cells, it is of great significance to develop integrated techniques with good biocompatibility and superior environmental tolerance.

The manuscript is well written, and most of its content is well described. The coated algal cells switch from the aerobic photosynthesis of oxygen to the hydrogenase-mediated production of hydrogen in air. This is a new attempt in this article. On the other hand, I found that some of the background information in the thesis is somewhat general, while some important points are not fully described. Thus, I recommend its publication for a Major revisions after the authors address the following questions:

Response: Many thanks for the reviewer's positive comments on our work. The following questions have been responded to point by point.

- 1) When exogenous materials are introduced into cells, their toxicities will increase significantly, resulting in cell function damage and even cell death. Can the authors clarify the

effects of the PPy/CaCO₃-coating methodology on the algal cells, to determine if there were any side effects related to toxicity?

Response: Many thanks for the comment. The effects of the PPy/CaCO₃-coating on the viability of the algal cells have been investigated by using fluorescein diacetate (FDA) as the indicator of cellular esterase activity and membrane integrity. It shows that the whole coating process including the absorption of Fe(III), the in situ polymerization of PPy and the crystallization of CaCO₃ imposed minimal damage on cell viability, and the viability of the engineered algal cells could maintain over 95% during the whole procedures.

The related data have been summarized in **Supplementary Figures 11 and 12d** of the original manuscript on Page 12 and 13 in Supporting Information.

2) Although the H₂ production increased in a short period of time, the rate of this system started to decrease due to destructed aggregation structures resulting from cell proliferation. how surface-augmented algal cell changes during the course of long working hours is unclear. The authors should further discuss them.

Response: In general, the outer CaCO₃ layer buffers excess protons in solution and limits cellular proliferation as indicated by the slower increase of chlorophyll content of PPy/CaCO₃-coated *Chlorella* cells (**Supplementary Figure 15**). As shown in **Fig. 5b**, by supplying extracellular electrons via periodical addition of TEOA at intervals of 6 days, the surface-augmented algal cells are able to maintain hydrogen production for at least 204 days, which we consider a relatively long period of time. To achieve this, cell viability is maintained by refreshing with TAP culture medium (as well as ascorbate, EY and TEOA) at 72, 126 and 168 days, as well as re-coating with CaCO₃ at these time points to regenerate an intact cell wall barrier with reduced macromolecular permeability (**Supplementary Figs. 10, 20 and 21**).

Our conclusion is that the surface-augmented algal cells remain functional provided the above procedures are followed. But without replenishment, the decrease in the hydrogen production rate is due to the normal apoptosis of the *Chlorella* cells.

We have clarified the text on Page 4 and Page 13 in the main text, and Page 14 in the Supporting Information as follows:

Page 4: "This was consistent with the well-maintained chlorophyll content of the PPy/CaCO₃-engineered cells observed over the same period (**Supplementary Fig. 15**)."

Page 13: "The decrease on the hydrogen production rate is due to normal apoptosis of the *Chlorella* cells."

SI Page 14: "The reduced rate of increase in chlorophyll content indicates that the rigid outer CaCO₃ layer limits proliferation of the coated *Chlorella* cells."

3) In Figure 5, EY and TEOA in the cell is not easily fixed because of the fluidity of the cytoplasm, making it difficult to bind the hydrogenase. Can the authors explain the proposed electron transfer mechanism more detailly?

Response: During the course of long-term hydrogen production, algal cells will gradually lose the viability and finally the photosystems are destroyed. The dead cells are more permeable, and the photosensitizer EY penetrates into the cytoplasm, which then directly arrives at hydrogenase via normal molecular diffusion as confirmed by the hydrogen production after the death of the coated algal cells. Alternatively, during the normal photosynthesis of the coated algal cell, we investigate the photochemical and non-photochemical processes by measuring the modulated chlorophyll fluorescence dynamics associated with the PPy/CaCO₃-coated cells in the presence or absence of EY and TEOA. As summarized in **Fig. 4**, we confirm that the extracellular electrons could also participate the direct and indirect processes of water photolysis. Therefore, two possible electron transfer pathways are proposed in the study. Pathway 1: electrons from EY and TEOA participate in photosynthesis, $PQ(H_2) \rightarrow b_6/f \rightarrow PC \rightarrow PSI \rightarrow Fd \rightarrow \text{hydrogenase}$, or pathway 2: direct electron transfer from EY to the hydrogenase via molecular diffusion.

According to the reviewer's suggestion, we have added the above explanations into the caption of **Fig. 4i** and **Fig. 5a** on Page 11 and 13, respectively.

Fig. 4i: "The extracellular electrons derived from EY and TEOA participate in the photosynthesis pathway ($PQ(H_2) \rightarrow b_6/f \rightarrow PC \rightarrow PSI \rightarrow Fd \rightarrow \text{hydrogenase}$), and both photolysis of water and endogenous organic matter contribute to photosynthetic hydrogen generation under hypoxic conditions."

Fig. 5a: "a, Schematic illustration of hydrogen production (paired cyan circles) arising from surface-augmented algal cells with the addition of EY and TEOA under hypoxic conditions. The electrons are directly transferred to hydrogenase via molecular diffusion."

Reviewer #3 (Remarks to the Author):

This study reports a nanobiohybrid cell system based on the interfacing of algal cells with a conductive PPy/CaCO₃ hybrid shell. The hybrid coating structurally stabilizes the microorganism, facilitates the onset and retention of a localized hypoxic micro-environment. Consequently, the coated algal cells switch from the aerobic photosynthesis of oxygen to the hydrogenase-mediated production of hydrogen in air. By addition of a photosensitizer and sacrificial electron donor in the extracellular environment, it can boost the light-dependent and light-independent reactions in the photosynthetic pathway, resulting in enhanced photosynthetic hydrogen production. Overall, it is an interesting work. I suggest some issues for the manuscript should be revised before being considered for publication.

Response: Many thanks for the reviewer's positive comments on our work.

1) Fig 1e is not very convincing. Also the absorption values are too high, you are probably saturating the detector. You need to reduce the concentration.

Response: Many thanks for the comment. By reducing the concentration of the tested samples,

we have repeated the measurement. The relevant experiment was refined, shown as below with a similar result, and the new data has been added as a revised **Fig. 1e**.

Revised Figure 1e. UV-Vis spectra of PPy, native and PPy-coated *Chlorella* cells.

2) In Fig 1g, it is not clear which section of the cell is imaged. Force measurements need to be done on a few different areas and average them out.

Response: Many thanks for the helpful comment. For the preparation of the samples for force measurements, the algal cells were dropped onto a silicon wafer, followed by the drying process to attach the cells to the substrate. The samples were then characterized by an atomic force microscope, where the random area (1 μm x 1 μm) of cellular surface on the top side was selected, imaged and then measured.

As suggested by the reviewer, to obtain more accurate statistics of the surface force values, three different areas (n = 3) were chosen, and the data shown as mean values \pm standard deviation in the revised **Fig. 1g**.

Revised Figure 1g. Plot of Young's modulus of native, PPy-coated and PPy/CaCO₃-coated *Chlorella* cell walls. Data are presented as mean values \pm SD, error bars indicate standard deviations (n = 3).

3) "nutrients in the external media remained accessible to the live *Chlorella* cells after coating of the cell wall"- Can you specify what are the nutrients and molecular sizes of these nutrient

molecules?

Response: The nutrients are trace metal ions in the TAP culture medium such as Mg^{2+} , K^+ , Mn^{2+} , Co^{2+} , as well as NH_4^+ and acetic acid. The molecular diameters for these species are less than 1 nm and can diffuse into the coated algal cells.

We have revised the text on Page 4 as follows:

“... suggesting that nutrients such as metal ions in the external media remained accessible to the live *Chlorella* cells after coating of the cell wall.”

4) In Fig S14, the coating improves thermal and acid stability. This is good results, but lacks explanation. The coating is just on the cell surface, how can it help maintain cell activities, which are happening inside the cells. In addition, wouldn't acid (pH 2) dissolves $CaCO_3$ coating? Even if not, the porosity would allow acid to diffuse into the cells.

Response: Many thanks for the insightful comments. The protective experiments of PPy/ $CaCO_3$ hybrid layer are designed to confirm the role of the coating materials on improving cellular resistance towards external stresses. For the thermal stability, the PPy/ $CaCO_3$ could stabilize the cell wall integrity under high temperatures, thus enhancing the resistance of the coated cells towards heat stresses - see also *ChemBioChem*, **2010**, 11, 2368-2373 and *Angew. Chem. Int. Ed*, **2013**, 52, 12279-12282.

For acid stability, we agree with the reviewer's comment. Actually, the outer $CaCO_3$ is also designed to serve as a sacrificial shell to buffer the excessive protons in the solution, which improves cellular activity under acidic conditions, especially given the pH sensitivity of the hydrogenases.

The following changes have been made to the revised main text:

Page 6: “.....the higher photosynthetic activity of hydrogen production observed in the presence of the outer $CaCO_3$ exoskeleton was due to buffering of protons in the solution (**Fig. 2d**).”

Page 7: “The $CaCO_3$ exoskeleton acts as a pH buffer against acidification of the environment.....”

Page 7: “The decrease of pH arises from the oxidation of added ascorbate and is buffered by the $CaCO_3$ shell, which also facilitates prolonged hydrogenase activity.

5) Beside the scheme in Fig 2A, the mechanism of O_2 consumption by Fe(III)-doped PPy is not clearly described and proved.

Response: The mechanism of O_2 consumption by Fe(III)-doped PPy is now included in **Supplementary Note 1**, on Page 22 of the Supporting Information. A reference to the mechanism has been added (*Int. J. Hydrogen Energy*, **2019**, 44, 17835-17844).

6) Fig 2d, why pH drop? It is not explained.

Response: Acidification of the culture medium is mainly caused by the oxidation of the added ascorbate, which is transformed into dehydroascorbic acid and then 2,3-diketogulonic acid (*Sci. Rep.*, **2021**, 11, 7417-7430). The related explanation has been added in the main text, on Page 6 and Page 7, which now read as follows:

Page 6: "During this period, oxidation of the substrate (ascorbate) within the native and PPy-coated algal cells gave rise to a decrease in pH from 7.2 to *ca.* 6.0 (**Fig. 2d**)".

Page 7: "The decrease of pH arises from the oxidation of added ascorbate and is buffered by the CaCO₃ shell"

7) In Fig 3d, the augmentation of photoelectron transfer should be also demonstrated using uncoated algae as a control.

Response: The control experiment where the native cells are added with EY and TEOA for photosynthetic hydrogen production is shown in **Supplementary Figure 18** in the Supporting Information. The results indicate that under the action of extracellular electrons, the hydrogen production of uncoated cells only increases by 9.87 μmol, compared to 51.71 μmol for PPy/CaCO₃-coated *Chlorella* cells.

8) The step-like pattern in H₂ production is probably an indication that the algae are not very active after a few days following new EY and TEOA and CaCO₃ addition. Can you reduce the interval of EY and TEOA and CaCO₃ renewal to optimize the H₂ production?

Response: Many thanks for the suggestion. Along with cellular metabolism, some unfavorable matter excreted from algal cells inhibits cell activity or hydrogenase, which results in the gradual decrease of photosynthetic hydrogen production. The hydrogen evolving kinetic study in **Fig. 5b** shows that the generated extracellular electrons not only increase the amount of photosynthetic hydrogen production, but also dramatically prolong the duration of hydrogen evolution by the modulation of photosynthesis and the direct supplying of exogenous electrons towards hydrogenase. For example, with the sustaining addition of EY and TEOA, hydrogen production period of algal cells is extended to 72 d, compared to 12 d for those without the treatment of EY and TEOA.

Since refreshing of culture medium and CaCO₃ shell would inevitably result in the exposure of algal cells to oxygen to some extent, thus imposing negative effects on hydrogen production, the interval of the addition of EY and TEOA and renewal of the CaCO₃ shell are optimized to achieve long-term hydrogen production.

9) Also it seems the coating is not very stable after a few days of operation. Can you provide an explanation and possible further improvements?

Response: Given the pH sensitivity of the hydrogenase under acid environment, the outer coating CaCO₃ shell is designed and constructed as a buffering agent to neutralize the excessive protons in the solution, maintaining the near-neutral conditions that are highly beneficial to the cell viability as well as hydrogenase activity. The CaCO₃ layer is inevitably decomposed

during this process. Consequently, in this study we strengthen and stabilize the CaCO_3 structure by additionally using CaCl_2 and Na_2CO_3 at the time points of 72 d, 126 d, 168 d and 204 d, which protects the algal cells and contributes to the long-term hydrogen production.

We have added the following in the main text on Page 7 to clarify this point:

“The CaCO_3 exoskeleton acts as a pH buffer against acidification of the environment.....”

Reviewers' Comments:

Reviewer #2:

Remarks to the Author:

In this revised manuscript, the authors have responded to the reviewers' questions very well. I recommend it for publication on Nature Communications.

Reviewer #3:

Remarks to the Author:

I am mostly OK with the responses. However, their responses to my concern regarding the thermal and acid protective role of the coating is still not clear to me. Even though the coating can stabilize cell wall under thermal stress, what happens to all the organic matters inside the cells? It is surprising that these organic matters like proteins do not unfold or deactivate under thermal or acidic conditions.

Reviewer #2 (Remarks to the Author):

In this revised manuscript, the authors have responded the reviewers' questions very well. I recommend it for publication on Nature Communications.

Response: We sincerely thank the reviewer's positive recommendation to our manuscript for publication.

Reviewer #3 (Remarks to the Author):

I am mostly OK with the responses. However, their responses to my concern regarding the thermal and acid protective role of the coating is still not clear to me. Even though the coating can stabilize cell wall under thermal stress, what happens to all the organic matters inside the cells? It is surprising that these organic matters like proteins do not unfold or deactivate under thermal or acidic conditions.

Response: Thank you very much for taking the time to review the revised manuscript again. Your careful evaluation on the thermal and acid protective role of the coating is much appreciated. We have carefully performed additional experiments on the protective role of the coating against time which are summarized as follows:

For the acid protectiveness experiment, when the native and PPy/CaCO₃-coated *Chlorella* cells are in the tested acid environment (hydrochloric acid, pH = 2, 2 mL and OD₇₅₀ of the cell = 3.0), the pH for the native cells remains close to 2, while for the PPy/CaCO₃-coated *Chlorella* cells, the outer CaCO₃ layer serves as a sacrificial shell to buffer the excessive protons in the solution, which raises the pH value to around 7.0 within 10 min (**Revised supplementary Fig. 13a**). Thus, cellular viability is significantly improved when the coated cells are exposed to an acidic environment.

For the thermal protectiveness, both the native and PPy/CaCO₃-coated cells gradually lose viability at the high temperature of 60 °C, due to the thermal-induced perforation of the cellular membrane and deactivation of functional proteins (**Revised supplementary Fig. 13b**). However, with the PPy/CaCO₃ hybrid layer, the integrity of the cellular membrane is better maintained, thus reducing the leakage of functional components from the inside of the cell (*ChemBioChem*, **2010**, 11, 2368-2373). This is indicated by the FDA experiment where the fluorescein is predominantly located inside the PPy/CaCO₃-coated algal cells (**Response Fig. 1**). Besides, the outer CaCO₃ shell may also absorb heat under a thermal environment, which releases the thermal stresses (*ChemBioChem*, **2010**, 11, 2368-2373) and retards the death of living cells within 60 min. But by extending the incubation time to 150 min, both the native and PPy/CaCO₃-coated cells lose their activity.

To avoid readers' confusion on this point, all the above discussions have been integrated into the caption of **Revised Supplementary Figure 13** (see below).

We hope the above responses clarify the point raised by the reviewer. We are not experimentally able to provide information on the precise structural changes to proteins inside the cells under thermal or acidic conditions but will undertake detailed biochemical analyses in future studies.

Revised Supplementary Figure 13. (a) Time-dependent measurements of pH for native and PPy/CaCO₃-coated *Chlorella* cells in acid solution (pH = 2). (b) Cell viability measurements of native and PPy/CaCO₃-coated *Chlorella* cells exposed to 60 °C for various times at neutral pH, at pH 2 for 60 min, or in the presence of toxic Ag nanoparticles at room temperature and neutral pH. For all experiments, the cells (OD₇₅₀ = 3.0) are immersed in 2 mL of DI water, except in the acid-protective experiment where the solution is replaced by hydrochloric acid (pH = 2). For the Ag nanoparticle-protective experiment, 100 μL of Ag nanoparticle (~0.75 A₅₂₀ units/mL, dispersed in sodium citrate) is added. Data are presented as mean values ± SD, error bars indicate standard deviations (n = 3). Coating with a PPy/CaCO₃ hybrid layer increases the integrity of the cellular membrane under adverse conditions, thus avoiding the leakage of cellular components under high temperature and increasing cell viability. In addition, the outer CaCO₃ layer serves as a sacrificial shell to buffer the acidic environment, which increases the pH to around 7.0 within 10 min, thus prolonging cellular viability.

Response Figure 1. Confocal fluorescence microscopy images showing the viability of native *Chlorella* cells (a) and PPy/CaCO₃-coated *Chlorella* cells (b) exposed to 60 °C for 30 min. Green fluorescence is from fluorescein in viable cells labelled with FDA, red fluorescence is from intracellular chlorophyll, and yellow fluorescence is from the overlay of the green and red fluorescence. Scale bar, 20 μm.

Reviewers' Comments:

Reviewer #3:

Remarks to the Author:

I am satisfied with the authors responses.

Reviewer #3 (Remarks to the Author):

I am satisfied with the authors' responses.

Response: Many thanks for the reviewer.